# RELITLRM: GENERATIVE RELIGHTABLE RADIANCE FOR LARGE RECONSTRUCTION MODELS

**Tianyuan Zhang**[1]   **Zhengfei Kuang**[2]   **Haian Jin**[3]   **Zexiang Xu**[4]   **Sai Bi**[4]   **Hao Tan**[4]
**He Zhang**[4]   **Yiwei Hu**[4]   **Milos Hasan**[4]   **William T. Freeman**[1]   **Kai Zhang**[4]*   **Fujun Luan**[4]*
[1]Massachusetts Institute of Technology    [2]Stanford University
[3]Cornell University    [4]Adobe Research

## ABSTRACT

We propose *RelitLRM*, a Large Reconstruction Model (LRM) for generating high-quality Gaussian splatting representations of 3D objects under novel illuminations from sparse (4-8) posed images captured under unknown static lighting. Unlike prior inverse rendering methods requiring dense captures and slow optimization, often causing artifacts like incorrect highlights or shadow baking, RelitLRM adopts a feed-forward transformer-based model with a novel combination of a geometry reconstructor and a relightable appearance generator based on diffusion. The model is trained end-to-end on synthetic multi-view renderings of objects under varying known illuminations. This architecture design enables to effectively decompose geometry and appearance, resolve the ambiguity between material and lighting, and capture the multi-modal distribution of shadows and specularity in the relit appearance. We show our sparse-view feed-forward RelitLRM offers competitive relighting results to state-of-the-art dense-view optimization-based baselines while being significantly faster. Our project page is available at: https://relit-lrm.github.io/.

## 1 INTRODUCTION

Reconstructing high-quality, relightable 3D objects from sparse images is a longstanding challenge in computer vision, with vital applications in gaming, digital content creation, and AR/VR. This problem is typically resolved by inverse rendering systems. However, most existing inverse rendering approaches suffer from limitations ranging from requiring dense captures under controlled lighting, slow per-scene optimization, using analytical BRDF hence not capable of modeling complex light transport, lacking data prior hence failure to resolve the ambiguity between shading and lighting in the case of static unknown lighting (Ramamoorthi & Hanrahan, 2001; Zhang et al., 2021b;a; 2022a; Jin et al., 2023; Kuang et al., 2024).

To overcome these limitations, we present *RelitLRM*, a generative Large Reconstruction Model (LRM) (Zhang et al., 2024a) that efficiently reconstructs high-quality, relightable 3D objects from as few as 4–8 posed images captured under unknown lighting conditions. Unlike traditional inverse rendering techniques that explicitly decompose appearance and shading, RelitLRM introduces an end-to-end relighting model directly controlled by environment maps. RelitLRM employs a feed-forward transformer-based architecture integrated with a denoising diffusion probabilistic model trained on a large-scale relighting dataset. This architecture models a multi-modal distribution of possible illumination decompositions, enabling deterministic reconstruction of object geometry and probabilistic relighting. In the case of glossy objects and challenging high-frequency novel lighting, the relit object appearance is inherently multi-modal with possibly many small specular regions; our relit appearance generator can well capture such high-frequency appearance, while regression-based approaches tend to be dominated by large smooth non-specular or weakly specular areas, and hence ignore the small strong specular highlights (see Fig. 5).

As shown in Fig. 1, our method can accurately reconstruct and relight the object's appearance under diverse lighting conditions and viewpoints, demonstrating its capability for photorealistic rendering

---

*Co-advise on the project

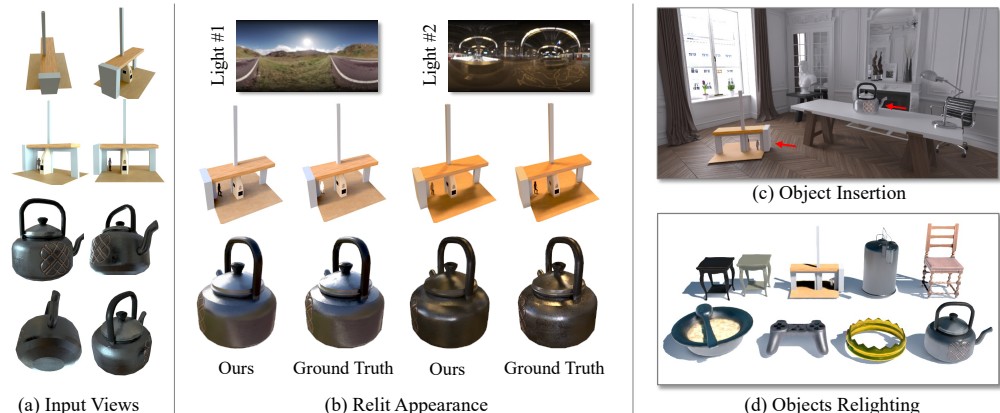

Figure 1: **Demonstration of *RelitLRM*'s relighting capabilities. (a)** Sparse posed images captured under unknown lighting conditions serve as the input for our method. **(b)** *RelitLRM* accurately reconstructs and relight the 3D object in the form of 3DGS under novel target lightings (either outdoor or indoor), with renderings closely matching the ground truth. **(c)** Object insertion into a virtual 3D scene. The red arrows indicate the objects relighted by *RelitLRM* using the scene illumination, demonstrating its ability to capture surrounding illumination and seamlessly harmonize with the scene background. **(d)** Objects relighting results showcase the robustness of our method under challenging lighting conditions, such as strong directional lighting. Our method faithfully models complex lighting effects, including removing shadow and highlights in input images and also casting strong shadows and glossy specular highlights under target lighting.

even with limited input views. By processing tokenized input images and target lighting conditions through transformer blocks, RelitLRM decodes 3D Gaussian (Kerbl et al., 2023) primitive parameters within approximately one second on a single A100 GPU. This approach disentangles geometry, appearance, and illumination, resolving material-lighting ambiguities more effectively than prior models and enabling photorealistic rendering from novel viewpoints under novel lightings.

We conduct extensive experiments on synthetic and real-world datasets to evaluate RelitLRM. The results demonstrate that our method matches state-of-the-art inverse rendering approaches while using significantly fewer input images and requiring much less processing time (seconds v.s. hours). Additionally, RelitLRM does not rely on fixed analytic reflectance models, allowing greater flexibility in handling diverse material properties. Its scalability and efficiency make it well-suited for real-world applications where flexibility in lighting conditions and rapid processing are essential. In summary, our key contributions to the field of relightable 3D reconstruction are:

- **Novel feed-forward generative 3D relighting architecture.** We end-to-end learn a generative 3D relightable reconstruction model with deterministic geometry reconstruction and probabilistic radiance generation. Our approach bypasses the explicit appearance decomposition and shading, and directly generates relighted radiance, producing realistic view-dependents appearances and self-shadows.

- **State-of-the-art performance with practical efficiency.** Trained on a large-scale 3D object dataset, RelitLRM matches or surpasses the performance of leading inverse rendering methods while using significantly fewer input images and less processing time, demonstrating superior generalization and practicality for real-world applications.

## 2 RELATED WORK

### 2.1 INVERSE RENDERING

Inverse rendering aims to recover intrinsic scene properties — geometry, material reflectance, and lighting — from images, enabling tasks like relighting and novel view synthesis. Traditional methods optimize through physically-based rendering equations (Kajiya, 1986) using dense image captures (typically 100–200 views) under controlled lighting (Debevec et al., 2000; Ramamoorthi &

Hanrahan, 2001; Marschner, 1998). Propelled by the recent advances in Neural Radiance Fields (NeRF) (Mildenhall et al., 2020) and 3D Gaussian Splatting (3DGS) (Kerbl et al., 2023), recent learning-based inverse rendering methods have emerged to decompose the scene properties with these neural representations (Bi et al., 2020b;a; Zhang et al., 2021a;b; Boss et al., 2021a;b; Luan et al., 2021; Srinivasan et al., 2021), by leveraging the differentiable rendering for their optimization and training (Boss et al., 2022; Zhang et al., 2022a;b; Kuang et al., 2022a;b; Jin et al., 2023; Shi et al., 2023; Yang et al., 2023; Zhang et al., 2023b; Gao et al., 2023; Liang et al., 2024; Jiang et al., 2024). However, these methods typically require dense inputs and computationally intensive optimization, limiting scalability and struggling with complex lighting and materials.

Our method addresses these challenges by efficiently reconstructing relightable 3D objects from sparse images (4–8 views) without per-scene optimization. Utilizing a transformer-based diffusion model integrated with the 3DGS (Kerbl et al., 2023), our method bypasses explicit appearance decomposition and shading, directly generates relighted radiance, enabling high-quality relighting and rendering under unknown lighting conditions with sparse views, and offering advantages in scalability and practicality

## 2.2 LARGE RECONSTRUCTION MODEL

Advancements in 3D reconstruction have been driven by neural implicit representations like NeRF (Mildenhall et al., 2020) and 3DGS (Kerbl et al., 2023), enabling high-quality novel view synthesis. Built upon it, the Large Reconstruction Model (LRM) series has further enhanced 3D reconstruction using transformer-based architectures for feed-forward processing. Notable models such as Single-view LRM (Hong et al.), Instant3D (Li et al., 2023), DMV3D (Xu et al.), PF-LRM (Wang et al., 2023), and GS-LRM (Zhang et al., 2024a), along with others (Wei et al., 2024; Xie et al., 2024; Anciukevičius et al., 2023; Szymanowicz et al., 2023), have introduced scalable methods for multi-view data, enabling detailed reconstructions even from sparse inputs.

Despite these improvements, integrating relighting capabilities remains challenging. Our approach, RelitLRM, extends the LRM series by incorporating relightable reconstruction, combining transformer-based architecture with neural implicit radiance representations within the 3DGS framework. This enables robust 3D reconstruction with relighting capabilities from sparse inputs, overcoming scalability and practicality issues of previous methods.

## 2.3 IMAGE-BASED RELIGHTING AND DIFFUSION MODELS

Diffusion models have become prominent in visual content generation, excelling in tasks like text-to-image synthesis and image editing (Ho et al., 2020; Nichol & Dhariwal, 2021). Models such as Stable Diffusion (Rombach et al., 2022) and ControlNet (Zhang et al., 2023a) adapt diffusion frameworks for controlled image synthesis. While primarily 2D, recent works like Zero-1-to-3 (Liu et al., 2023b) demonstrate diffusion models' latent understanding of 3D structures, relevant for view synthesis and relighting.

Single-image relighting is challenging due to the limited view of input data. Early neural network-based approaches (Ren et al., 2015; Xu et al., 2018) laid the groundwork, with recent portrait-focused methods (Sun et al., 2019; Zhou et al., 2019; Pandey et al., 2021; Mei et al., 2024; Ren et al., 2024) enhancing realism and lighting control. However, these methods are often domain-specific or rely on explicit scene decomposition, limiting adaptability. Recent methods like Neural Gaffer (Jin et al., 2024), DiLightNet (Zeng et al., 2024), and IC-Light (Zhang et al., 2024b) leverage diffusion models for precise lighting control, achieving high-quality relighting. However, All theses works are constrained to single-view inputs.

Our approach integrates generative relighting and 3D reconstruction into an end-to-end transformer-based architecture. It produce high-quality renderings of objects from novel viewpoints and under arbitrary lighting, given sparse input images.

## 3 METHOD

Our RelitLRM is built on top of the prior GS-LRM (Zhang et al., 2024a) model, but differs in crucial aspects: 1) we aim for relightability of the output assets; 2) to achieve this, we propose a

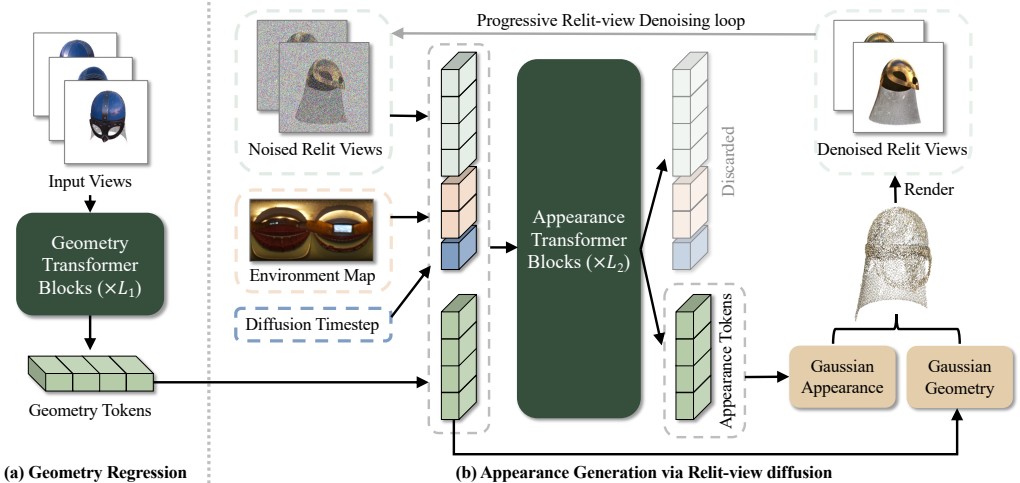

Figure 2: **Overview of *RelitLRM* for sparse-view relightable 3D reconstruction.** Our pipeline consists of a geometry regressor and a relit appearance generator, both implemented as transformer blocks and trained jointly end-to-end. We implicitly bake the relit appearance generation in relit-view diffusion process. During inference, we first extract geometry tokens from sparse input images and regress geometry parameters for per-pixel 3D Gaussians (3DGS). Conditioning on novel target lighting and extracted geometry features, we denoise the relit views by first predicting a 3DGS appearance, then render it (along with 3DGS geometry that stays fixed in the diffusion denoising loop) into the denoising viewpoints. This iterative process produces the relit 3DGS radiance as a byproduct while denoising the relit views. The generative appearance and deterministic geometry blocks are trained end-to-end, ensuring scalability.

novel generative relighting radiance component to replace GS-LRM's deterministic appearance part. In this section, we first briefly review the model architecture of GS-LRM, then describe in detail our relit 3D appearance generation algorithm based on relit-view diffusion.

## 3.1 REVIEW OF GS-LRM MODEL

GS-LRM (Zhang et al., 2024a) uses a transformer model for predicting 3DGS parameters from sparse posed images. They demonstrated instant photo-realistic 3D reconstruction results. We provide an overview of their method below.

Given a set of $N$ images $I_i \in \mathcal{R}^{H \times W \times 3}, i = 1, 2, ..., N$ and corresponding camera Plücker rays $\mathbf{P}_i$, GS-LRM concatenates $I_i, \mathbf{P}_i$ channel-wise first, then converts the concatenated feature map into tokens using patch size $p$. The multi-view tokens are then processed by a sequence of $L_1$ transformer blocks for predicting 3DGS parameters $\mathbf{G}_{ij}$: one 3DGS at each input view pixel location.

$$\{\boldsymbol{T}_{ij}\}_{j=1,2,...,HW/p^2} = \text{Linear}\big(\text{Patchify}_p\big(\text{Concat}(I_i, \boldsymbol{P}_i)\big)\big), \tag{1}$$

$$\{\boldsymbol{T}_{ij}\}^0 = \{\boldsymbol{T}_{ij}\}, \tag{2}$$

$$\{\boldsymbol{T}_{ij}\}^l = \text{TransformerBlock}^l(\{\boldsymbol{T}_{ij}\}^{l-1}), \quad l = 1, 2, ..., L_1, \tag{3}$$

$$\{\mathbf{G}_{ij}\} = \text{Linear}(\{\boldsymbol{T}_{ij}\}^{L_1}). \tag{4}$$

The whole model is trained end-to-end using novel-view rendering loss on multi-view posed images.

## 3.2 GENERATING RELIT 3DGS APPEARANCE VIA RELIT VIEW DIFFUSION

The relighting procedure involves the removal of source lighting from input images, and relighting the objects under target lighting, which are inherently ambiguous in the case of sparse view observations and unknown lighting conditions, especially in regions of shadows and specularity. Moreover, when relighting glossy objects using high-frequency novel illuminations, the relit object appearance is inherently multi-modal with possibly many small specular regions; regression-based approaches tend to be dominated by large diffuse or weakly specular areas, ignoring the small strong specu-

lar highlights. Hence, to boost relighting quality, we need a generative model to better handle the uncertainty in relighting and multi-modality in relit radiance.

We design a new approach to generate relit 3DGS appearance under a novel lighting. Related to prior work (Xu et al.; Szymanowicz et al., 2023; Anciukevičius et al., 2023), we perform diffusion, in which we force a 3D representation as a bottleneck. However, the key algorithmic innovation is that our appearance predictor is a relighting diffusion model, conditioned on the novel target illumination and the predicted geometry features. The relit-view denoiser uses $x_0$ prediction objective (Ramesh et al., 2022); at each denoising step, it outputs the denoised relit views by predicting and then rendering the relit 3D Gaussian representation.

Formally, we first predict 3DGS geometry parameters $\{\mathbf{G}^{\text{geom}}_{ij}\}$ using the same approach (see Eq. 1,2,3,4) as in GS-LRM (Zhang et al., 2024a); we also obtain the last transformer block's output $\{\mathbf{T}^{\text{geom}}_{ij}\}$ of this geometry network to use as input to our relit-view denoiser. To produce the relit 3DGS apperance under a target lighting, we then concatenate the noisy relit view tokens, target lighting tokens, denoising timestep token, and predicted 3DGS geometry tokens, passing them through a second transformer stack with $L_2$ layers. The process is described as follows:

$$\{\mathbf{T}^{\text{all}}_i\}^0 = \left\{ \{\mathbf{T}^{\text{noised-relit-views}}_{ij}\}^{L_1} \oplus \{\mathbf{T}^{\text{light}}_{ij}\} \oplus \{\mathbf{T}^{\text{geom}}_{ij}\} \oplus \{\mathbf{T}^{\text{timestep}}\} \right\}, \tag{5}$$

$$\{\mathbf{T}^{\text{all}}_i\}^l = \text{TransformerBlock}^{L_1+l}(\{\mathbf{T}^{\text{all}}_i\}^{l-1}), \quad l = 1, 2, \ldots, L_2. \tag{6}$$

The output tokens $\{\mathbf{T}^{\text{geom}}_{ij}\}^{L_2}$ are used to decode the appearance parameters of the 3D Gaussians, while the other tokens, e.g., $\{\mathbf{T}^{\text{noised-relit-views}}_{ij}\}^{L_2}$ are discarded on the output end. In detail, we predict the 3DGS spherical harmonics (SH) coefficients via a linear layer:

$$\{\mathbf{G}^{\text{color}}_{ij}\} = \text{Linear}\left(\{\mathbf{T}^{\text{geom}}_{ij}\}^{L_2}\right), \tag{7}$$

where $\mathbf{G}^{\text{color}}_{ij} \in \mathbb{R}^{75p^2}$ represents the 4-th order SH coefficients for the predicted per-pixel 3DGS in a $p$x$p$ patch.

At each denoising step, we output the denoised relit views (using $x_0$ prediction objective) by rendering the predicted 3DGS representation:

$$\{\mathbf{I}^{\text{denoised-relit-views}}_i\} = \text{Render}\left(\{\mathbf{G}^{\text{geom}}_{ij}\}, \{\mathbf{G}^{\text{color}}_{ij}\}\right). \tag{8}$$

The interleaving of 3DGS appearance prediction and rendering in the relit-view denoising process allow us to generate a relit 3DGS appearance as a side product during inference time. We output the 3DGS at final denoising step as our model's final output: a 3DGS relit by target lighting.

**Tokenizing HDR environment maps.** To make it easier for networks to ingest HDR environment maps $E \in \mathcal{R}^{H_e \times W_e \times 3}$, we convert each $E$ into two feature maps through two different tone-mappers: one $E_1$ emphasizing dark regions and one $E_2$ for bright regions. We then concatenate $E_1, E_2$ with the ray direction $D$, creating a 9-channel feature for each environment map pixel. Then we patchify it into light tokens, $\{\mathbf{T}^{\text{light}}_{ij}\}_{j=1,2,\ldots,H_e W_e/p_e^2}$, with a MLP. See Appendix 9 for details.

## 3.3 TRAINING SUPERVISION

In each training iteration, we sample a sparse set of input images for an object under a random lighting and another set of images of the same object under different lighting (along with its ground-truth environment map). The input posed images are first passed through the geometry transformer to extract object geometry features. As described in Sec. 3.2, we then perform relighting diffusion with the $x_0$ prediction objective; our denoiser is a transformer network translating predicted 3DGS geometry features into 3DGS appearance features, conditioning on the sampled novel lighting and noised multi-view images under this novel lighting (along with diffusion timestep).

We apply $\ell_2$ and perceptual loss (Chen & Koltun, 2017) to the renderings at both the diffusion viewpoints and another two novel viewpoints in the same novel lighting. The deterministic geometry predictor and probabilistic relit view denoiser are trained jointly end-to-end from scratch.

## 3.4 SAMPLING RELIT RADIANCE

At inference time, we first reconstruct 3DGS geometry from the user-provided sparse images. Then we sample a relit radiance in the form of 3DGS spherical harmonics given a target illumination. Our

relit radiance is implictly generated by sampling the relit-view diffusion model. We use the DDIM sampler (Song et al., 2020) and classifier-free guidance technique (Ho & Salimans, 2022). Since the denoised relit views are rendered from the 3DGS geometry and predicted relit radiance at each denosing step, the relit-view denoising process also results in a chain of generated relit radiances. We only keep the relit radiance at last denoising step as our final predicted 3DGS appearance.

## 4 DATASET AND IMPLEMENTATION

**Training dataset.** Our training dataset is constructed from a combination of 800K objects sourced from Objaverse (Deitke et al., 2023) and 210K synthetic objects from Zeroverse (Xie et al., 2024). The inclusion of Zeroverse data, which features a higher proportion of concave objects, assists in training the model to handle complex shadowing effects. Additionally, We also modify the metallic properties and roughness of some Zeroverse objects to introduce more specular surfaces, adding variety and challenge to the dataset.

For lighting diversity, we gathered over 8,000 HDR environment maps from multiple sources, including Polyhaven[1], Laval Indoor (Gardner et al., 2017), Laval Outdoor (Hold-Geoffroy et al., 2019), internal datasets, and a selection of randomly generated Gaussian blobs. We augmented these maps with random horizontal rotations and flips, increasing their number 16-fold. Each object is rendered from 10 to 32 randomly selected viewpoints under 2 to 12 different lighting conditions.

**Model details.** Our model comprises 24 layers of transformer blocks for the geometry reconstruction stage and an additional 8 layers for the appearance diffusion (denoising) stage, with a hidden dimension of 1024 for the transformers and 4096 for the MLPs, totaling approximately 0.4 billion trainable parameters. We employ Pre-Layer Normalization (Xiong et al., 2020) and GeLU activations (Hendrycks & Gimpel, 2016).

Input images and environment maps are tokenized using an $8 \times 8$ patch size, while denoising views are tokenized with a $16 \times 16$ patch size to optimize computational efficiency. Tokenization involves a simple reshape operation followed by a linear layer, with separate weights for the input and target image tokenizers. The diffusion timestep embedding is processed via an MLP similar to Peebles & Xie (2023), is appended to the input token set to the diffusion transformer.

**Training details.** The initial training phase employs four input views, four target denoising views (under target lighting, used for computing the diffusion loss), and two additional supervision views (under target lighting), all at a resolution of $256 \times 256$, with the environment map set to $128 \times 256$. The model is trained with a batch size of 512 for 80K iterations, introducing the perceptual loss after the first 5K iterations to enhance training stability. Following this pretraining at the 256-resolution, we fine-tune the model for a larger context by increasing to six input views and six denosing target views at a higher resolution of $512 \times 512$. This fine-tuning expands the context window to up to 31K tokens. For diffusion training, we discretize the noise into 1,000 timesteps, adhering to the method described in Ho et al. (2020), with a variance schedule that linearly increases from 0.00085 to 0.0120. To enable classifier-free guidance, environment map tokens are randomly masked to zero with a probability of 0.1 during training. For more details, please refer to Appendix.

**Sampling details.** We performed an ablation study on the key hyperparameters, specifically the number of denoising steps and the classifier-free guidance (CFG) weight, as detailed in Table 6. We found our model can produce realistic and diverse results with five sampling steps, and CFG of 3.0 gives the best result for our res-256 model.

## 5 EXPERIMENTS

**We refer the readers to our project page for video results and interactive visualizations.**

---

[1] https://polyhaven.com/hdris

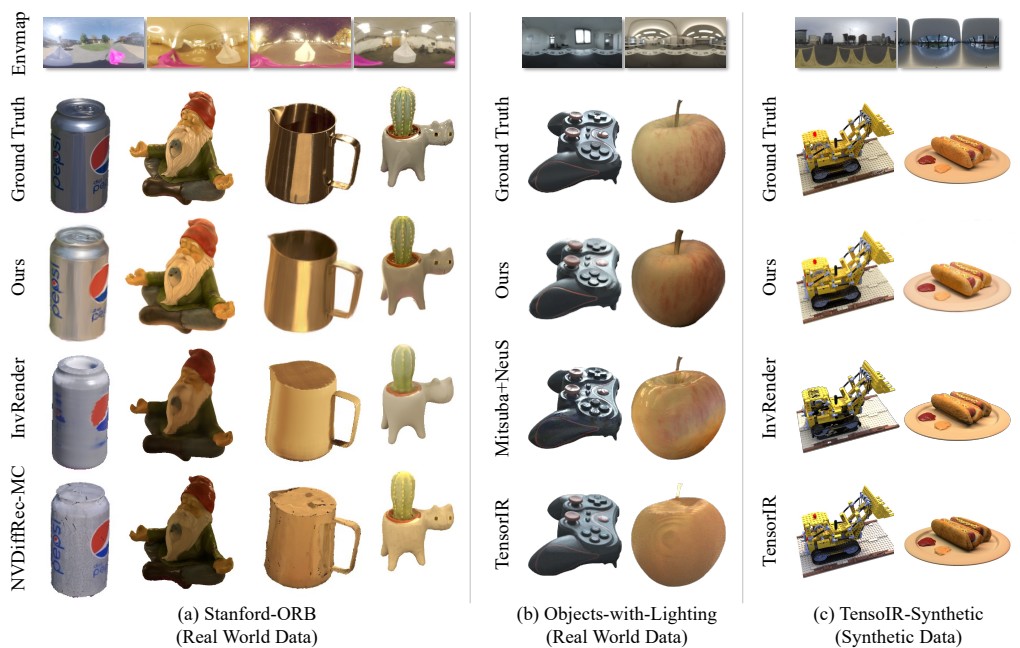

Figure 3: **Comparison with optimization-based inverse rendering baselines** on STANFORD-ORB (Kuang et al., 2024), OBJECTS-WITH-LIGHTING (Ummenhofer et al., 2024), and TENSOIR-SYNTHETIC (Jin et al., 2023) datasets. (a) On STANFORD-ORB, our method captures realistic specular highlights and geometric details, outperforming NVDiffRec-MC(Hasselgren et al., 2022) and InvRender (Zhang et al., 2022b). (b) On OBJECTS-WITH-LIGHTING, our results closely match the ground truth, while baselines show over-specularity or artifacts. (c) For TENSOIR-SYNTHETIC, our method achieves comparable relighting with significantly fewer views. Notably, our method requires only 6 to 8 views and completes relighting in 2-3 seconds, while baselines need over 50 views and hours of processing. No color rescaling is performed for visualization.

## 5.1 COMPARISON WITH OPTIMIZATION-BASED METHODS

Current optimization-based inverse rendering algorithms achieve state-of-the-art 3D relighting results, but typically require dense view captures (over 40 views) and long optimization times, often taking several hours, using traditional rendering techniques like rasterization, path tracing, or neural rendering.

We evaluate our method against these approaches on three publicly available datasets: STANFORD-ORB (Kuang et al., 2024), OBJECTS-WITH-LIGHTING (Ummenhofer et al., 2024), and TENSOIR-SYNTHETIC (Jin et al., 2023). The STANFORD-ORB dataset comprises 14 objects captured under three lighting conditions, with around 60 training views and 10 test views per lighting setup per object. The OBJECTS-WITH-LIGHTING dataset contains 7 objects with dense views captured under one training lighting condition and 3 views for two additional lighting conditions for testing. The TENSOIR-SYNTHETIC dataset consists of 4 objects with 100 training views under one lighting condition and 200 test views for each of five lighting conditions.

Our method requires only **six input views** for STANFORD-ORB and OBJECTS-WITH-LIGHTING and **eight input views** for TENSOIR-SYNTHETIC, completing the entire reconstruction and relighting process in just **2 to 3 seconds**. In contrast, state-of-the-art optimization-based methods typically rely on all available training views (around 60 views per object for STANFORD-ORB, 42-67 views per object for OBJECTS-WITH-LIGHTING, and 100 views per object for TENSOIR-SYNTHETIC) and take several hours to optimize. Despite using significantly fewer input views and processing time, our method achieves comparable reconstruction and relighting results, as demonstrated in Table 1 and Table 2, showcasing its remarkable efficiency. Qualitative comparisons in Figure 3 further highlight our method's superior performance across various scenarios. Optimization-based methods struggle with high uncertainties when decomposing object appearances under input lighting and relighting, leading to baked highlights (Column-6 in Figure 3) and poor specular reflections (Columns

1 and 3). In contrast, our generative approach handles these challenges effectively. Relighted images is rescaled when evaluating, following each benchmark's protocol for Table 1, 2.

Table 1: **Real world experiments**. We compare our method with state-of-the-art optimization-based approaches on the STANFORD-ORB (Kuang et al., 2024) and OBJECTS-WITH-LIGHTING (Ummenhofer et al., 2024) datasets. Our method offers competitive relighting results, but uses only six input views and runs within seconds on a single A100 GPU, whereas other methods use ∼60 views for STANFORD-ORB and 42 to 67 views for Objects-With-Lighting, taking hours to complete.

| Method | STANFORD-ORB | | | | OBJECTS-WITH-LIGHTING | | | # Input views | Runtime |
|---|---|---|---|---|---|---|---|---|---|
| | PSNR-H ↑ | PSNR-L ↑ | SSIM ↑ | LPIPS ↓ | PSNR ↑ | SSIM ↑ | LPIPS ↓ | | |
| Ours | 24.67 | 31.52 | 0.969 | 0.032 | 23.08 | 0.79 | 0.284 | **6** | **∼2 seconds** |
| Neural-PBIR | **26.01** | **33.26** | **0.979** | **0.023** | N/A | N/A | N/A | 42-67 | hours |
| Mitsuba+Neus | N/A | N/A | N/A | N/A | **26.24** | **0.84** | **0.227** | 42-67 | hours |
| IllumiNeRF | 25.56 | 32.74 | 0.976 | 0.027 | N/A | N/A | N/A | 42-67 | hours |
| NVDIFFREC-MC | 24.43 | 31.60 | 0.972 | 0.036 | 20.24 | 0.73 | 0.393 | 42-67 | hours |
| InvRender | 23.76 | 30.83 | 0.970 | 0.046 | 23.45 | 0.77 | 0.374 | 42-67 | hours |
| TensoIR | N/A | N/A | N/A | N/A | 24.15 | 0.77 | 0.378 | 42-67 | hours |
| NeRFactor | 23.54 | 30.38 | 0.969 | 0.048 | 20.62 | 0.72 | 0.486 | 42-67 | hours |
| NVDIFFREC | 22.91 | 29.72 | 0.963 | 0.039 | 22.60 | 0.72 | 0.406 | 42-67 | hours |
| PhySG | 21.81 | 28.11 | 0.960 | 0.055 | 22.77 | 0.82 | 0.375 | 42-67 | hours |

Table 2: **Comparison with dense-view optimization-based methods on TensoIR-Synthetic.** This dataset from (Jin et al., 2023) includes 4 scenes, each having 100 training views and 5 lighting conditions with 200 test views. We report per-scene and average metrics. Baselines use 100 input views; our method uses only eight. * indicates rescaling in pixel space instead of albedo.

| Method | Armadillo | | Ficus | | Hotdog | | Lego | | Average | | | # Input views | Runtime |
|---|---|---|---|---|---|---|---|---|---|---|---|---|---|
| | PSNR | SSIM | PSNR | SSIM | PSNR | SSIM | PSNR | SSIM | PSNR | SSIM | LPIPS | | |
| Ours* | 33.25 | 0.966 | 26.18 | 0.952 | **29.32** | 0.938 | 27.44 | 0.886 | 29.05 | 0.936 | 0.082 | **8** | **∼3 second** |
| FlashCache | 34.81 | 0.959 | 26.29 | **0.960** | 29.24 | **0.941** | 28.56 | 0.917 | **29.72** | 0.944 | 0.080 | 100 | hours |
| IllumiNeRF* | **35.36** | 0.974 | **27.36** | 0.959 | 27.95 | 0.939 | 28.14 | 0.915 | 29.71 | **0.947** | 0.072 | 100 | hours |
| TensoIR* | 34.75 | 0.974 | 25.66 | 0.950 | 28.78 | 0.932 | **29.37** | **0.932** | 29.64 | **0.947** | **0.068** | 100 | hours |
| TensoIR | 34.50 | **0.975** | 24.30 | 0.947 | 27.93 | 0.933 | 27.60 | 0.922 | 28.58 | 0.944 | 0.081 | 100 | hours |
| InvRender | 27.81 | 0.949 | 20.33 | 0.895 | 27.63 | 0.928 | 20.12 | 0.832 | 23.97 | 0.901 | 0.101 | 100 | hours |
| NeRFactor | 26.89 | 0.944 | 20.68 | 0.907 | 22.71 | 0.914 | 23.25 | 0.865 | 23.38 | 0.908 | 0.131 | 100 | hours |

## 5.2 COMPARISON WITH DATA-DRIVEN BASED RELIGHTING METHODS

We also compare our method with state-of-the-art image relighting approaches (Jin et al., 2024; Zeng et al., 2024), which fine-tune text-to-image diffusion models for single-image relighting. These methods, however, only support relighting from the input views and do not handle novel view relighting. Real-world benchmarks like STANFORD-ORB and OBJECTS-WITH-LIGHTING focus on novel view relighting, making these methods unsuitable for direct evaluation on those datasets.

To ensure a fair comparison, we construct a held-out set from filtered Objaverse (unseen during training), comprising 7 highly specular objects and 6 concave objects with shadows. Each object is rendered under five environment maps with four viewpoints, resulting in a total of 260 images for evaluation. We compute PSNR, SSIM, and LPIPS metrics, after applying per-channel RGB scaling to align predictions with the ground truth, following (Jin et al., 2024). Our method outperforms the baselines, showing better image quality metrics as in Table 3.

## 5.3 OBJECT SCENE INSERTION

RelitLRM's capability to accurately reconstruct and relight 3D objects makes it particularly effective for object scene insertion tasks, where the goal is to seamlessly integrate objects into existing environments, as demonstrated in Fig. 1(c) and Fig. 1(d). In Fig. 1(c), we demonstrate the insertion of two objects into an indoor living-room scene, where they harmonize naturally with the surrounding lighting and shadows. In Fig. 1(d), we present that eight objects are relit under strong directional sunlight with our method, showcasing their ability to cast realistic shadows and exhibit accurate specular highlights, demonstrating RelitLRM's effectiveness in handling diverse lighting conditions.

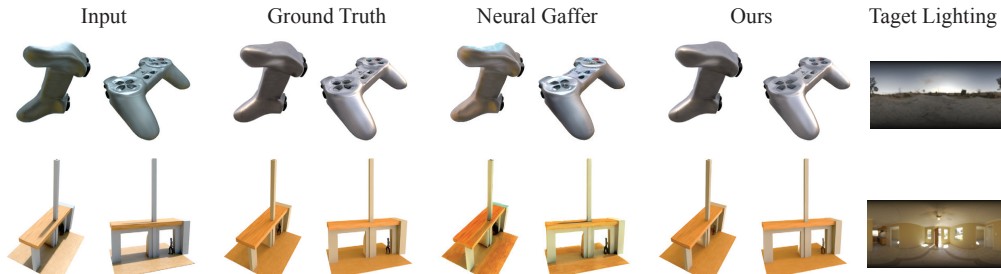

Figure 4: **Comparison with image-based relighting baseline** on our held-out evaluation set shows that our model produces better visual quality, with improved shadow removal and highlight. Our model processes four input images jointly, while the baseline relights each image independently.

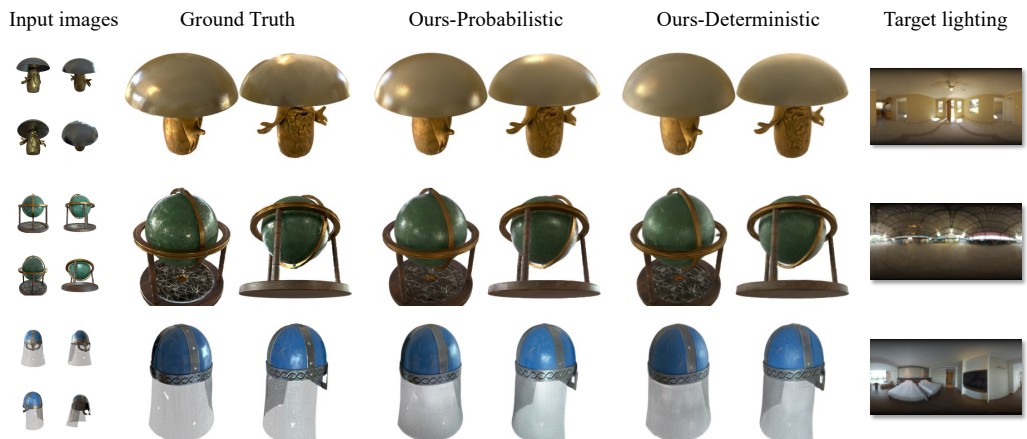

Figure 5: **Our probabilistic design yields significantly better results on specular highlights compared to the deterministic counterpart.** The radiance function of specular objects under challenging lighting is highly multi-modal with long tails. Our denoising diffusion approach models this distribution more effectively, while the deterministic design fails to mode such complex distribution and produce overly smooth specular highlights.

## 5.4  ABLATION STUDY

**Deterministic vs. probabilistic relighting.**  Decomposing object's appearance from input lighting and relighting is inherently ambiguous. Moreover, the radiance function of object under high-frequency lighting is extremely complex and highly multi-modal, e.g. containing multiple small sharp highlights. To address this, our model reconstructs geometry deterministically while relighting the object's radiance probabilistically using a diffusion approach. To evaluate this design, we introduce a deterministic baseline that relights objects directly, without the diffusion step, using the same architecture, amount of parameters, and training iterations. We assess both models on our hold-out validation set, with metrics presented in Table 4 and visual comparisons in Fig. 5. Although the quantitative results are similar, the probabilistic approach significantly improves visual quality for specular objects. This supports the idea that our probabilistic design captures the complex multi-modal radiance function more effectively than the deterministic counterpart, which tends to produce over-smooth results.

**Effects of more input views.**  We evaluate our model with gradually increased number of input images up to 16 (70K input tokens) on our held-out set and TensoIR-Synthetic as shown in Table 5. We observe a positive correlation between our relighting quality and the number of input images.

**Hyper-parameters in diffusion.**  In our model, the number of denoising views is independent of the number of input views. After all denoising steps, the virtual denoising views are discard, leaving only the 3D Gaussians for relighting at arbitrary views. A key question is how the denoising views affect the results. Additionally, two other parameters significantly impact relighting performance:

Table 3: **Comparison with image-based re-lighting models on held-out validation set.** The prediction is scaled per-channel to align with the target following (Jin et al., 2024). The PSNR is only computed in masked foreground regions. Our method takes four input views, and baseline methods take each view independently.

| | PSNR ↑ | SSIM ↑ | LPIPS ↓ | NFE | Parameters |
|---|---|---|---|---|---|
| Ours | **28.50** | **0.93** | **0.058** | 5 | 0.4B |
| NeuralGaffer | 24.51 | 0.90 | 0.072 | 50 | 0.9B |
| DiLightNet | 20.02 | 0.85 | 0.132 | 20 | 0.9B |

Table 4: **Ablation on probabilistic design**. We design a deterministic counterpart by removing the noisy image tokens and keeping the model architecture and training configuration the same. We compare our probabilistic design with the deterministic counterpart on held-out evaluation set. See Figure 5 for visual results.

| | Input Views | PSNR ↑ | SSIM ↑ | LPIPS ↓ |
|---|---|---|---|---|
| Ours-Probabilistic | 4 | **27.63** | **0.922** | **0.064** |
| Ours-Deterministic | 4 | 27.10 | 0.912 | 0.076 |

Table 5: **Impact of more input views**: We evaluate the Res-512 model by incrementally adding more input views and compare its performance on our hold-out evaluation set and TENSOIR-SYNTHETIC. The Res-512 model is fine-tuned from the Res-256 model using six input views and six denoising views, and fixed through this evaluation.

| HOLDOUT VALIDATION SET | | | | TENSOIR-SYNTHETIC | | | |
|---|---|---|---|---|---|---|---|
| Input Views | PSNR ↑ | SSIM ↑ | LPIPS ↓ | Input Views | PSNR ↑ | SSIM ↑ | LPIPS ↓ |
| 2 | 23.58 | 0.896 | 0.113 | 4 | 27.83 | 0.926 | 0.091 |
| 4 | 26.14 | 0.915 | 0.086 | 8 | 29.04 | 0.936 | 0.082 |
| 6 | 27.71 | 0.926 | 0.072 | 12 | 29.27 | 0.938 | 0.081 |
| 8 | **28.08** | **0.929** | **0.070** | 16 | **29.31** | **0.938** | **0.080** |

Table 6: **Ablation on the parameter space of diffusion** with four input views at $256 \times 256$ resolution in our holdout evaluation set. Default setting is marked as `gray`.

(a) **Number of denoising views**.

| views | PSNR ↑ | SSIM ↑ | LPIPS ↓ |
|---|---|---|---|
| 1 | 27.05 | 0.913 | 0.075 |
| 2 | 26.87 | 0.913 | 0.0745 |
| 4 | 27.63 | 0.922 | 0.064 |

(b) **Classifier-free guidance**.

| CFG | PSNR ↑ | SSIM ↑ | LPIPS ↓ |
|---|---|---|---|
| 1.0 | 27.00 | 0.915 | 0.072 |
| 3.0 | 27.63 | 0.922 | 0.064 |
| 6.0 | 23.64 | 0.891 | 0.0974 |

(c) **Number of denoising steps**.

| steps | PSNR ↑ | SSIM ↑ | LPIPS ↓ |
|---|---|---|---|
| 2 | 23.81 | 0.884 | 0.096 |
| 5 | 27.63 | 0.922 | 0.064 |
| 10 | 27.00 | 0.915 | 0.073 |

the classifier free guidance weight and number of denoising steps. We present qualitative ablation results in Table 6. For classifier-free guidance, we visualize its effect in Fig. 6, where the unconditional predictions looks like relighting under average lighting in our dataset, and higher CFG weights making the object appear more specular.

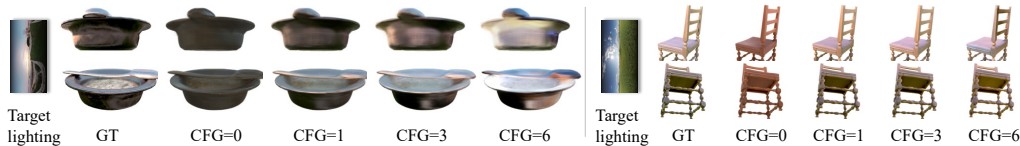

Target lighting | GT | CFG=0 | CFG=1 | CFG=3 | CFG=6     Target lighting | GT | CFG=0 | CFG=1 | CFG=3 | CFG=6

Figure 6: **Effect of classifier-free-guidance (CFG).** We show two novel-view relighting results with different CFG weight. Zero weight denotes unconditional relighting, which resembles relighting with average dataset lighting. Higher CFG weight makes the object more specular.

## 5.5 LIMITATIONS

Despite our model reconstructs and relights objects from sparse-view images, it requires camera parameters for the input views, which can be impractical in some cases. Also our model does not support material editing applications, due to lack of explicit appearance decompositions. Although the model improves output quality with more input images, performance saturates around 16 views (as shown in Table 5). Improving scalability with additional input views and higher resolutions remains an area for future exploration. Additionally, our model uses environment maps as the target lighting representation, which cannot accurately represent near-field lighting.

## 6 CONCLUSION

we presented RelitLRM, a generative Large Reconstruction Model for reconstructing high-quality, relightable 3D objects from sparse input images using a diffusion transformer. Unlike traditional methods, our approach bypasses explicit appearance decomposition and shading, instead generating relighted radiance directly through a deterministic geometry reconstructor followed by a probabilistic appearance generator based on relit-view diffusion. Trained on extensive relighting datasets, RelitLRM captures complex lighting effects such as shadows and specularity, often surpassing state-of-the-art baselines while requiring significantly fewer input images. Moreover, it achieves this in just 2–3 seconds, compared to the hours needed by per-scene optimization methods, demonstrating its remarkable efficiency and potential for real-world applications in relightable 3D reconstruction.

**Ethics Statement.** This work proposes a large reconstruction model with a generative relightable radiance component, which can be used to reconstruct relightable 3D assets from sparse images. The proposed model tries best to preserve the identity of objects in the captured images in the reconstruction process; hence it might be used for reconstructing 3D from images with humans or commerical IPs.

**Reproducibility Statement.** We have included enough details in this main paper (Sec. 4) and appendix (Sec. A.1) to ensure reproducibility, which includes our training and testing data, model architecture, training setup, etc. We are also happy to address any questions regarding the implementation details of our paper, if any.

**Acknowledgements.** This work was started when Tianyuan Zhang, Zhengfei Kuang and Haian Jin were intern at Adobe Research. This research was in part sponsored by the Department of the Air Force Artificial Intelligence Accelerator and was accomplished under Cooperative Agreement Number FA8750-19-2-1000, the NSF PHY-2019786 (The NSF AI Institute for Artificial Intelligence and Fundamental Interactions, http://iaifi.org/), and NSF CIF 1955864 (Occlusion and Directional Resolution in Computational Imaging).

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

# A APPENDIX

## A.1 PSEUDO CODE

---

**Algorithm 1** RelitLRM pseudo code.

---

```
# one training iter of RelitLRM
# Input list:
# x_inp: [b, n, h, w, 3], n is number of views; h and w are the height and width;
# x_inp_ray_o, x_inp_ray_d: both [b, n, h, w, 3]. per-pixel ray direction and ray
      origins for input images.
# target_relit_image: [b, n1, h, w, 3], n1 is the number of denosing views.
# extra_train_image: [b, n2, h, w], extra ground truth relighted image for
      supervision.
# envmap: [b, 3, h_e, w_e]. target lighting environment map
# Output: training loss

# Step-1: geometry token extraction from input images
x_inp = concat([x_inp, x_inp.ray_plucker], dim=1) # [b, n, h, w, 9]
x_inp = conv1(x_inp, out=d, kernel=8, stride=8) # patchfy: [b, n, h/8, w/8, d]
x_inp = x_inp.reshape(b, -1, d) # [b, n_inp_tokens, d]
x_inp_geo = transformer1(LN(x_inp))

# step-2: tokenize envmap
e_darker = log(envmap) / max(log(envmap))
e_brighter = HLG(envmap) # hybrid log-gamma transfer function
x_light = concat([e_darker, e_brighter, envmap.ray_d], dim=-1) # [b, h_e, w_e, 9]
x_light = conv2(x_light, out=d, kernel=8, stride=8)
x_light = x_light.reshape(b, -1, d) # [b, n_light_token, d]
x_light = linear(GeLU(x_light), out=d) # [b, n_light_token, d]

# step-3: add noise to target_relit_image
t = randint(0, 1000, (x_inp.shape[0], )) # [b,]
alpha_cumprod = ddpm_scheduler(t)
noise = randn_like(target_relit_image)
x_t_relight = target_relit_image * sqrt(alpha_cumprod) + noise * sqrt(1 -
      alpha_cumprod)
t_token = TimeStepEmbedder(t) # [b, 1, d]

# step-4: tokenize noisy relit images
# concate with plucker ray of denosing views to [b, n', h, w, 9]
x_t_relight = concate([x_t_relight, target_relit_image.ray_plucker])
x_t_relight = conv3(x_t_relight, out=d, kernel=16, stride=16)
x_t_relight = x_t_relight.reshape(b, -1, d) # [b, n_relight_token, d]

# step-5: feed every tokens to the denosing transformer
# everything as tokens
x_all = concat([x_inp_geo, x_light, x_t_relight, t_token], dim=1)
# [b, n_inp_tokens + n_light_tokens + n_relight_tokens + 1, d]
x_all = transformer2(LN(x_all))
x_inp_radiance, _ = x_all.split(x_inp_geo.shape[1], dim=1) # [b, n_inp_token, d]

# step-6: decode pixel-algined 3D Gaussian parameters
# geometry parameters
x_geo = x_inp_geo.reshape(b, n, h//8, w//8, d)
x_geo = deconv(LN(x_geo), out=9, kernel=8, stride=8) # unpatchify to pixel-aligned GS
      output: [b, n, h, w, 9]
x_geo = x_geo.reshape(b, -1, 9) # all geometry params [b, n * h * w, 9]
# apperance_parameters, spherical-harmonics up to order of 4
x_inp_radiance = x_inp_radiance.reshape(b, n, h//8, w//8, d)
sh_weight = deconv(LN(x_inp_radiance), out=75, kernel=8, stride=8) # 75-dim sh

# GS parameterization
distance, scaling, rotation, opacity = x_geo.split([1, 3, 4, 1], dim=-1)
w = sigmoid(distance)
xyz = x_inp_ray_o + x_inp_ray_d * (near * (1 - w) + far * w)
scaling = min(exp(scaling - 2.3), 0.3)
rotation = rotation / rotation.norm(dim=-1, keepdim=True)
opacity = sigmoid(opacity - 2.0)

# step-7: compute training loss
3dgs = (xyz, rgb, scaling, sh_weight, rotation, opacity)

pred_x0, pred_extra_view = Render(3dgs, [target_relit_image.camera, extra_train_image
      .camera])

l2_loss = L2([pred_x0, pred_extra_view], [target_relit_image, extra_train_image])
lpips_loss = LPIPS([pred_x0, pred_extra_view], [target_relit_image, extra_train_image
      ])
loss = l2_loss + 0.5 * lpips_loss
```

---

A.2 MODEL DETAILS

**Tokenizing HDR environment maps.** To make it easier for networks to ingest HDR environment maps $E \in \mathcal{R}^{H_e \times W_e \times 3}$, we convert each $E$ into two feature maps through two different tone-mappers: one $E_1$ emphasizing dark regions and one $E_2$ for bright regions.

$$E_1 = \frac{\log_{10}(E)}{\max(\log_{10}(E))}, \quad E_2 = \mathrm{HLG}(E), \tag{9}$$

where $\mathrm{HLG}(\cdot)$ represents the hybrid log-gamma transfer function. We then concatenate $E_1, E_2$ with the ray direction $D$, creating a 9-channel feature for each environment map pixel. Then we patchify it and apply a two layer MLP with to turn it into light tokens, $\{\boldsymbol{T}_{ij}^{\mathrm{light}}\}_{j=1,2,\ldots,H_e W_e/p_e^2}$.

A.3 IMAGE AND LIGHTING PREPROCESSING

To address scale ambiguity in relighting, we normalize both the rendered images and the environment map lighting in the dataset. For a rectangular target environment map, we compute its total energy, weighted by the sine of each pixel's elevation angle. We then normalize the environment map to ensure the weighted total energy matches a predefined constant. The same scaling factor is applied to the HDR rendered images. The detailed process is outlined in pseudo code 2.

---

**Algorithm 2** Normalize environment map and rendered images .

```
# normalize the rendered images and environment maps
# Input list:
# x: [n, h, w, 3], n is number of views rendered for this scene under this lighting.
    h is image widht and w is image height.
# envmap: [h_e, w_e, 3]. environment map used to render this scene.
# constant_energy: pre-defined total energy. precomputed as dataset median number
# output list: x_normalized, envmap_normalized

# Step-1: compute weighted total energy of the environment map
H, W = envmap.shape[:2]
theta = linspace(0.5, H - 0.5, H) / H * np.pi
sin_theta = sin(theta)
envmap_weighted = envmap * sin_theta[:, None, None]
weighted_energy = sum(envmap_weighted)

# step-2: compute scaling factor
scale = constant_energy / weighted_energy

# step-3: normalize envmap
envmap = envmap * scale

# step-4: normalize and tonemap rendered images
x = x * scale # [n, h, w, 3]
x_flatten = x.reshape(-1, 3) # [n_total_pixel, 3]
x_min, x_max = percentile(x_flatten, [1, 99])
x = (x - x_min) / (x_max - x_min)
x = x.clip(0, 1)
# tonemapping
x = x ** (1.0 / 2.2)

return x, envmap
```

---

A.4 TRAINING HYPERPARAMETERS

We begin by training the model from scratch with four input views, four denoising views, and two additional supervision views under target lighting, at a resolution of $256 \times 256$. The model is trained for 80K iterations with a batch size of 512, using the AdamW optimizer with a peak learning rate of $4e - 4$ and a weight decay of 0.05. The $\beta_1, \beta_2$ are set to 0.9 and 0.95 respectively. We use 2000 iterations of warmup and start to introduce perceptual loss after 5000 iterations for training stability. We then finetune the model at a resolution of $512 \times 512$ with six input views, six denoising views, and two supervision views, increasing the number of input tokens to 31K. This finetuning runs for 20K iterations with a batch size of 128, using the AdamW optimizer with a reduced peak learning rate of $4e - 5$ and 1000 warmup steps. Throughout training, we apply gradient clipping at 1.0 and skip steps where the gradient norm exceeds 20.0.

## A.5 TRAINING COMPUTE

Our transformer model has a total of 400M parameters. Initially, we train it at a resolution of $256 \times 256$ over 270B input tokens, which requires four days on 32 NVIDIA A100 GPUs (40GB VRAM each). Subsequently, we fine-tune the model at a higher resolution of $512 \times 512$ with six input views, processing an additional 79B input tokens over two days.

## A.6 GEOMETRY EVALUATIONS

We evaluate the geometry quality of our method on the STANFORD-ORB(Kuang et al., 2024) dataset using our Res-512 model with six input views. For evaluation, we filter out Gaussian points with opacity below $0.05$ and randomly sample 30,000 points. The Chamfer distance is computed against the scanned ground-truth mesh using the official Stanford-ORB script, with results scaled by $2 \times 10^3$, as per the benchmark protocol. Table 7 summarizes the geometry and relighting results. Our method achieves the best geometry score while requiring only six input images and seconds of processing time, in contrast to other methods that rely on dense view inputs and require hours for final relighting.

Table 7: **Geometry evaluation on real world dataset**. We evaluate the geometry quality of our method against state-of-the-art optimization-based relighting approaches on the STANFORD-ORB (Kuang et al., 2024) dataset. Our method achieves superior geometry quality with comparable relighting results, using only six input views and running within seconds on a single A100 GPU. In contrast, competing methods require approximately 60 views for STANFORD-ORB.

| Method | GEOMETRY Chamfer distance ↓ | NOVEL VIEW RELIGHTING PSNR-H ↑ | PSNR-L | # Input views | Runtime |
|---|---|---|---|---|---|
| Ours | **0.39** | 24.67 | 31.52 | **6** | **∼2 seconds** |
| Neural-PBIR | 0.43 | **26.01** | **33.26** | ∼60 | hours |
| IllumiNeRF | N/A | 25.56 | 32.74 | ∼60 | hours |
| NVDIFFREC-MC | 0.51 | 24.43 | 31.60 | ∼60 | hours |
| InvRender | 0.44 | 23.76 | 30.83 | ∼60 | hours |
| NeRFactor | 9.53 | 23.54 | 30.38 | ∼60 | hours |
| NVDIFFREC | 0.62 | 22.91 | 29.72 | ∼60 | hours |
| PhySG | 9.28 | 21.81 | 28.11 | ∼60 | hours |

## A.7 ATTENTION MAP VISUALIZATION

We visualize the attention map between appearance tokens and environment map tokens. Our relighting diffusion transformer has eight transformer layers, has typically five inference steps. We visualize the attention map of the last transformer layer of the last denoising timestep.

Each transformer layer employs multi-head self-attention with 16 heads, each of dimension 64, resulting in a total hidden dimension of 1024. For visualization, we concatenate the key and value vectors from all heads and compute attention using the aggregated keys and values.

We use our Res-256 model at four-view input setup. for visualization. We show visualized results for two objects, each with two input lighting setups and two target lightings. For each input setup, we visualize the attention map for the image patch at the center of the first and second input image. For each target lighting, we horizontally shift it by $1/4, 1/2, 3/4$ to create four target environment map. For each target lighting, we horizontally shift the environment map by $1/4, 1/2,$ and $3/4,$ creating four variations per lighting. In total, we visualize $2 \times 2 \times 2 \times 4 = 32$ attention maps per object. The results are shown in Figures 7 and 8.

For the visualized attention, we summarize some empirical patterns, though not super consistent.

- **Lighting rotation consistency**. The attention map between appearance token and environment map tokens is quite consistent to rotations of the environment map. Note that environment map is tokenized by concating the color and ray direction for each pixel. This consistency implies a strong dependence on directional cues in environment map.

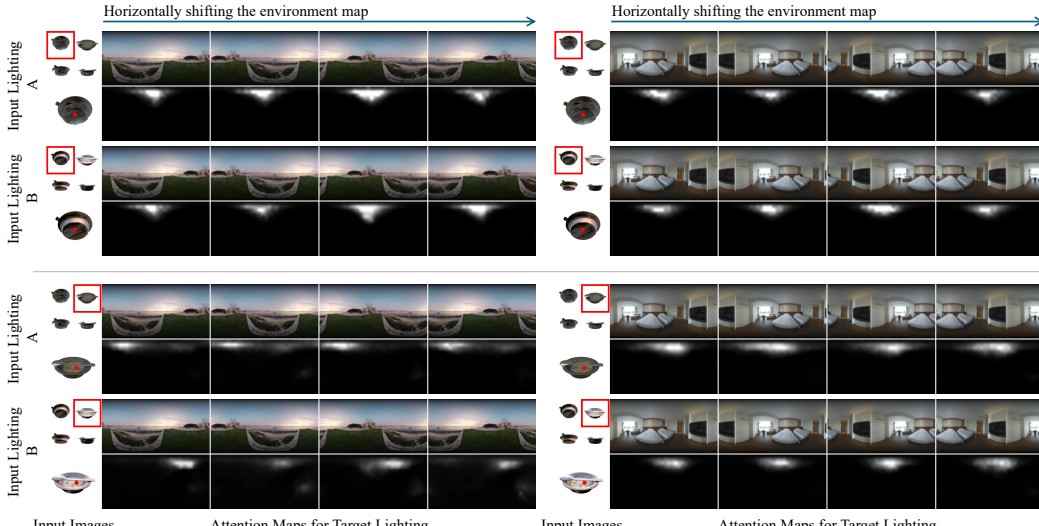

Figure 7: **Attention maps between appearance tokens and environment map tokens.** The red dot in the input images indicates the corresponding appearance token. Visualized attention maps are shown for two target lighting, each horizontally shifted by $1/4$, $1/2$, and $3/4$, creating four variations per lighting. Results from two sets of input images are displayed, captured under different lighting conditions but from the same set of viewpoints. This figure features a specular object.

- **Input lighting stability**. While less consistent than rotation, attention maps are relatively stable across changes in input lighting. This might suggest, that the attention map learned something about the appearance models of the objects.

- Contrary to intuition, attention maps for specular objects (Figure 7) are not noticeably sharper or sparser compared to those for diffuse objects, as seen in Figure 8.

- Highlights in attention maps do not consistently align with directions where target lighting most affects object appearance. For instance, in Figure 7 (first row), appearance tokens corresponding to the object's bottom aggregate lighting from the top of the environment map, contrary to expectation.

## A.8 SCALING THE NUMBER OF INPUT VIEWS DURING INFERENCE

Our model demonstrates context-length generalization, where increasing the number of input views leads to improved results (quantitatively shown in Table 5). To better illustrate this effect, we incrementally visualize relighting results for the "Lego" object with 4, 8, and 12 input views in Figure 9. The highlighted region clearly shows improvement from 4 to 12 views due to enhanced scene coverage.

We observed that increasing input views tends to results in brighter outputs. To investigate this phenomenon, we conducted an experiment (Figure 10) using an 8-view input but selectively visualizing Gaussians predicted from only the first few views, without modifying the transformer's forward pass.

To interpret these results, recall our model details: each input image predicts pixel-aligned 3D Gaussians, and we directly concatenate predicted Gaussians from all input views as the final output. Thus, increasing the number of input views leads to more overlapping Gaussians.

Key observations from Figure 10: In overlapping regions between views, some Gaussians appear masked out. However, remember that we predict a Gaussian at each pixel, and these white regions are indeed visible to the first four viewpoints, these Gaussians in the white region is not explicitly masked out, but learned a low opacity scores. We also visualize the predicted Gaussians from the first 1, 2, 3 input views to better understand this effect(still the 8 input view setting.) in Figure-12.

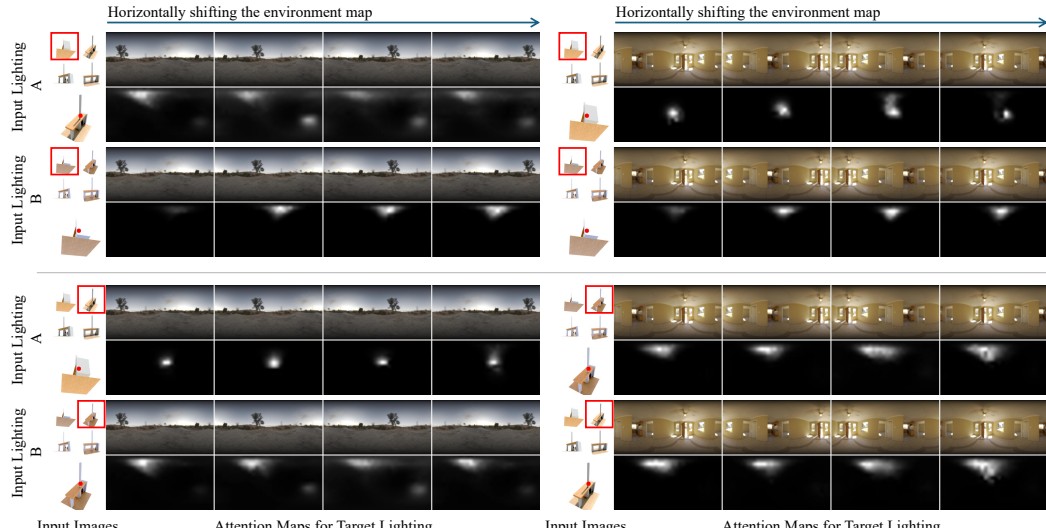

Figure 8: **Attention maps between appearance tokens and environment map tokens.** The red dot in the input images indicates the corresponding appearance token. Visualized attention maps are shown for two target lighting, each horizontally shifted by $1/4$, $1/2$, and $3/4$, creating four variations per lighting. Results from two sets of input images are displayed, captured under different lighting conditions but from the same set of viewpoints. This figure features a diffuse object.

Our hypothesis: We conjecture that with more and more input views, there will be more and more floaters with low opacity around the objects, especially in overlapping regions between input views. And these floaters makes the object appear brighter and brighter. We also want to clarify that these floaters might not be meaningless, they should have learned to also add some finner details to the final results.

To validate this hypothesis, we visualized the results after pruning Gaussians with opacity scores below 0.25. For 8-view and 12-view inputs, this reduced the object's brightness while maintaining visually high-quality renderings, as in the updated version of the Figure 9 (last two columns of the bottom half of the figure). However, pruning also removed some high-frequency details, suggesting that these low-opacity floaters are a mechanism learned by the model to enhance detail with more input views, with a side-effect of changing object's brightness.

This observation also highlights a key challenge for scaling feed-forward 3D models like GS-LRM and RelitLRM to handle a large number of input views (e.g., over 100). Improved methods for aggregating predictions across viewpoints, or entirely new paradigms, may be required to address this issue effectively.

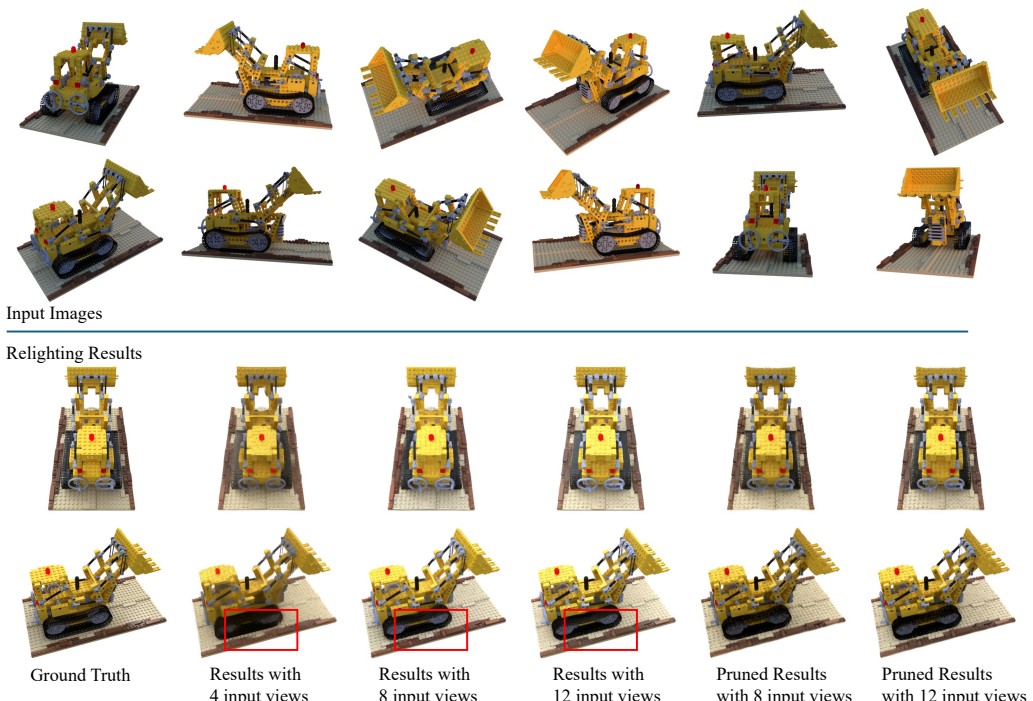

Figure 9: **Results with more input views.** The top part of the figure displays 12 input images, with the first four images in the first row corresponding to the 4-view setup. The bottom part shows results with progressively more input views (4, 8, and 12 views). In the region highlighted by the red rectangle, the 12-view result demonstrates significant improvement over the 4-view result due to better coverage of the scene from input images. The last two column of the bottom part shows relighting results after pruning Gaussian with opacity below 0.25.

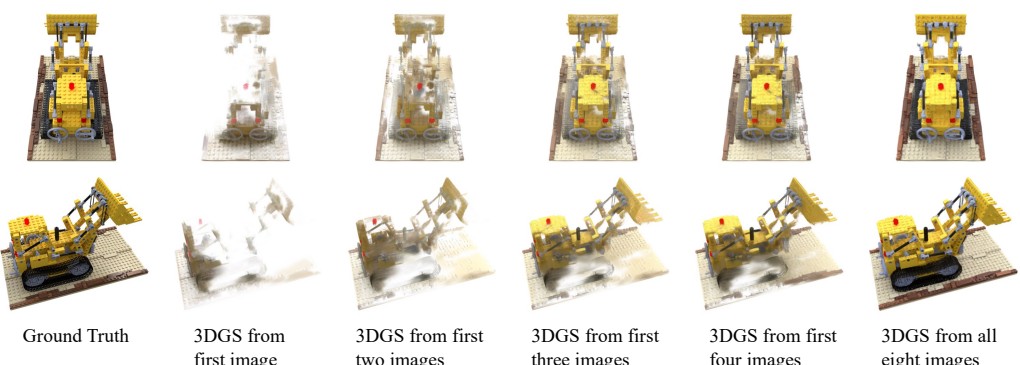

Figure 10: We feed eight input images to our model, instead of concating predicted pixel-aligned Gaussians from all viewpoints, we gradually visualize the predicted Gaussian from the first few viewpoints.

## A.9   ADDITIONAL RESULTS FOR FIGURE 3

For quantitative evaluation in the relighting task, the standard practice is to channel-wise rescale relighted images before computing metrics. However, the qualitative results shown in the main paper did not apply this rescaling. To address this, we provide additional rescaled results for the "Pepsi" object from the Stanford-ORB dataset in Figure 7, where channel-wise scaling is applied to better match the ground-truth image.

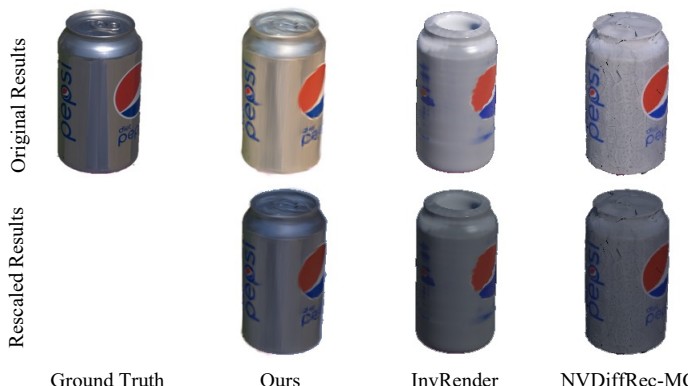

Figure 11: **Results after rescaling for "Pepsi".** We present the original results from the first column of Figure 3 (first column) alongside rescaled versions for better visualization. Rescaling is applied channel-wise, using a factor computed from the median values of the ground truth and the corresponding relighting results.

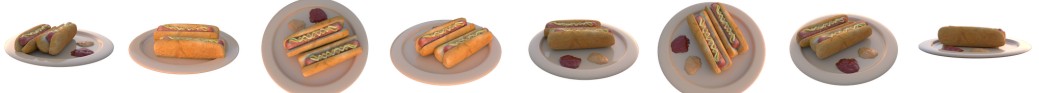

Figure 12: **Input images of our method for the "Hotdog".** While baseline methods require 100 dense-view input images, our approach uses only these eight views. These input images shows shadows at the bottom part of the hotdog, which the baseline methods shown in Figure 3 fail to remove effectively.

Additionally, Figure 8 shows input views used by our method for the "Hotdog" object (Figure 3(c)). Our sparse-view approach effectively removes shadows from input images, outperforming baseline methods like InvRender and TensoIR, which produce overly dark results at the object's bottom due to poor shadow removal. While our shadows are softer and sometimes less accurate compared to ray-marching-based methods (e.g., TensoIR), our method achieves these results without explicit shadow-specific inductive biases.

## A.10 ARCHITECTURE ANALYSIS

Our model processes input tokens from multiple modalities, including posed input images, target environment maps, noisy relit images, and denoising timestep tokens. We designed our architecture as a uni-stream model, where we use lightweight MLP layers to map each modality (environment map tokens, input image tokens, noise image tokens, and timestep tokens) into a shared feature space. Then, full self-attention is applied to extract and aggregate features across all tokens.

A natural question is why we chose self-attention instead of cross-attention. To address this concern, we offer the following rationale:

- Simplicity: Self-attention simplifies the architecture by eliminating the need for modality-specific encoders (e.g., separate encoders for images and environment maps) and a decoder. Traditional approaches introduce complexity, such as determining how to allocate parameters across modalities. In contrast, we use lightweight MLPs to project tokens into a shared space, followed by self-attention, allowing the model to learn how to allocate capacity dynamically across modalities and tasks.

- Our design is inspired by large-scale vision-language models like LLava Liu et al. (2023a; 2024) and PaliGemma Beyer et al. (2024), which treat image and text tokens equivalently after projection via lightweight MLPs, followed by self-attention layers. Similarly, our model treats all input modalities equivalently after projection, avoiding complex inductive biases.

To compare the impact of self-attention versus cross-attention, we replaced the self-attention layers in the relighting transformer with cross-attention layers while keeping the number of trainable parameters constant. Both models were trained with 4 input views at Res-256 resolution for 80k iterations using the same setup and data. The results of this comparison on the Stanford-ORB, Objects-with-Lighting, and Held-out Evaluation Sets are shown in Table 8:

Table 8: Self-Attention vs. Cross-Attention Performance Comparison

| Dataset | Method | PSNR-H ↑ | PSNR-L ↑ | SSIM ↑ |
|---|---|---|---|---|
| Stanford-ORB | Cross-Attention | 21.06 | 26.70 | 0.943 |
|  | Self-Attention | **22.97** | **29.42** | **0.967** |
| Objects-with-Lighting | Cross-Attention | N/A | 13.46 | 0.491 |
|  | Self-Attention | N/A | **20.62** | **0.756** |
| Held-out Evaluation Set | Cross-Attention | N/A | 26.34 | 0.906 |
|  | Self-Attention | N/A | **27.63** | **0.922** |

