# OpenReview forum: "RelitLRM: Generative Relightable Radiance for Large Reconstruction Models"
_ICLR.cc/2025/Conference — ICLR 2025 Spotlight_

### Official Review · Reviewer_Sueo · 2024-11-01

**Soundness:** 3
**Presentation:** 3
**Contribution:** 2
**Rating:** 6
**Confidence:** 3

**Summary:**

This paper proposes a method trained end-to-end on synthetic multi-view renderings of objects under varied, known illuminations. The approach is able to generate high-quality Gaussian splatting representations of 3D objects under novel illuminations from sparse input images (4-8 views) captured in unknown, static lighting conditions.

**Strengths:**

- The method models geometry and rendering independently, akin to NeRF’s modeling architecture. It first constructs geometry from multi-view images, then generates the rendering by combining geometry features with environmental maps.
- The method is practical as it only requires 4-8 views as input. It is able to effectively reconstruct relightable objects without per-scene optimization.
- On both synthetic and real-world datasets, the method offers competitive relighting results, requiring much fewer input views than other methods.

**Weaknesses:**

- In the paper, the authors claim that performance plateaus at around 16 views. However, as shown in Table 5, there is only a marginal improvement in image quality on the TensoIR-Synthetic dataset when increasing from 8 views to 16 views.
- The novelty of the approach needs clearer articulation. The authors state that their method differs from Neural Gaffer (Jin et al., 2024) by not being limited to single-view inputs. However, this advantage seems to stem from the use of GS-LRM (Zhang et al., 2024a). It is important to clarify how their application of diffusion for relighting distinguishes their method from existing techniques.
- This method separates geometry from rendering, but the paper does not show results of the decomposed geometry. It is unclear how good or bad the quality of the reconstructed geometries is.

**Questions:**

- In the paper, the authors provide the rendering performance. However, I cannot find the training time. Please provide more specifics on the training setup, such as the number of GPUs used and total training hours.
- A more thorough analyais and discussion of why performance plateaus between 8 to 16 views would enhance the paper's quality.
- Please provide quantitative evaluations of the extracted geometries.
- In the object insertion image (Figure 1(c)), how is illumination in the scene accounted for? Did you sample on the position of the object to capture the surrounding environment and incorporate the environment map into your model? Additionally, how do you account for the indirect light from another object produced by your RelitLRM in the scene?
- The method still takes 2 to 3 seconds to render. In contrast, the geometry and textures obtained from other methods can be rendered using many real-time rendering techniques. Moreover, in current industrial applications, it is challenging to abandon triangle meshes and textures. Therefore, this method cannot be considered a viable solution for 3D digital assets. However, if this approach could be extended to scene relighting, its applicability could be significantly broadened.
- Is the number of Gaussians predicted by the model sufficient for capturing different topologies and structures across diverse objects?

---

> ### Author Response · Authors · 2024-11-20
> **First rebuttal For Reviewer Sueo. Part-1.  Performance with more views.  & Geometry Evaluation & Object Insertion & Training setup & Number of Gaussians**
>
> Due to the 5000-character limit per reply, I have split the rebuttal into multiple parts. This is Part 1 of the first round of rebuttal with Reviewer Sueo.
>
> We thanks the detailed comments and appreciation of our work from Reviewer Sueo, here we address your comments
>
> ## Performance with more views.
> **Reviewer Sueo**:
> *“In the paper, the authors claim that performance plateaus at around 16 views. However, as shown in Table 5, there is only a marginal improvement in image quality on the TensoIR-Synthetic dataset when increasing from 8 views to 16 views.”*
>
> *“A more thorough analyais and discussion of why performance plateaus between 8 to 16 views would enhance the paper's quality.”*
>
> Yes, we agree, going from 8 views to 16 views only yields marginal improvements on TensoIR-Synthetic.  While the gains are modest, we shouldn’t take these marginal improvements for granted, because our model is only trained with six input views. During inference, using 12 or 16 views introduces a considerable distribution shift by doubling or tripling the input tokens; it's already non-trivial to produce results without catastrophic degradation.
>
> To make the model really excel at handling dense views, models need to be trained with more views. And training with more views, e.g. 16 and 32 input views poses two challenges:
> 1. Huge computational cost with current transformers. (for the attention module, computational cost increases quadratically with the number of input tokens). So maybe other more efficient architecture needs to be explored, like linear attention alternatives. (personally, I think the task of 3D reconstruction needs matching between all image patches, e.g. correspondence matching, and linear attention cannot do global random matching between tokens well).
> 2. Need a more effective and efficient way of merging per-view 3D Gaussian. Our current approach predicts one 3D Gaussian per pixel and concatenates them directly for output. This linear increase in Gaussians with views is computationally expensive and produce too much redundancy for dense-view scenarios.
>
> Currently, optimization-based methods like structure-from-motion, 3D Gaussians and other inverse rendering methods can process thousands of images; However, data-driven feedforward methods can only take sparse views. Developing scalable models for dense-view 3D feedforward methods is an open and impactful area for future research.
>
> We show the visual results of using 4, 8, 12 input views for the lego in the Figure-9 of the updated Appendix. Most of the improvements come from more accurate textures in regions covered insufficiently with fewer inputs. And I think more such visualization definitely helps reader in understanding our model, and we will provide more.
>
>
> ## Evaluation of Geometry
> **Reviewer Sueo**:
> *“This method separates geometry from rendering, but the paper does not show results of the decomposed geometry. It is unclear how good or bad the quality of the reconstructed geometries is.”*
>
> We appreciate this suggestion, and we will show the evaluation of geometry in the Stanford-ORB dataset with chamfer distance between our reconstructed Gaussians and scanned ground-truth meshes. I will update these results in two days.
>
> ## About object insertion image
> **Reviewer Sueo**:
> *In the object insertion image (Figure 1(c)), how is illumination in the scene accounted for? Did you sample on the position of the object to capture the surrounding environment and incorporate the environment map into your model? Additionally, how do you account for the indirect light from another object produced by your RelitLRM in the scene?*
>
> In Figure 1(c), we sample an environment map at the position of the object. To cast shadows from the inserted objects into the scene, we reconstruct a coarse mesh of the object from the output 3D Gaussians using Poisson surface reconstruction and render the shadows with Blender.
>
> Indirect lighting from other inserted objects is not accounted for in Figures 1(c) and 1(d).
>
>
> ## Training setup
> **Reviewer Sueo**:
> *“Questions about training time, training setup”*
>
> Thanks for pointing this out. This question is also raised by other two reviewers. We already added it in A.4 of the Appendix.
>
> We train our model with 32 A100GPUs(40G VRAM). For the Res-256 Model, we train for 4 days, and for the Res512 model, we finetune for another 2 days.
>
> ## Number of Gaussians
> **Reviewer Sueo**:
> *Is the number of Gaussians predicted by the model sufficient for capturing different topologies and structures across diverse objects?*
>
> We predict one pixel-aligned Guaissian from each pixel. For the Res512 model with four input images, we will output over one million Gaussians. We believe this is enough for objects with different topologies and structures.

---

> ### Author Response · Authors · 2024-11-20
> **First rebuttal For Reviewer Sueo. Part-2.   Clearer articulation of novelty**
>
> **Reviewer Sueo:**
> *“The novelty of the approach needs clearer articulation. The authors state that their method differs from Neural Gaffer (Jin et al., 2024) by not being limited to single-view inputs. However, this advantage seems to stem from the use of GS-LRM (Zhang et al., 2024a). It is important to clarify how their application of diffusion for relighting distinguishes their method from existing techniques”*
>
> Thank you for raising this point. We would like to clarify the distinctions between our method and GS-LRM, particularly regarding the application of diffusion for relighting. While our approach builds upon recent advances in data-driven feedforward 3D models, including LRM, LGM and GS-LRM, there are several key differences that make our work unique:
>
> **Probabilistic v.s. Deterministic Design.**
> A fundamental distinction is that our method is probabilistic, while GS-LRM is deterministic, and This difference leads to non-trivial performance improvements, particularly in handling specular objects.  Let me elaborate more on this:
>
> Normally when one wants to build up GS-LRM for relighting, it will design a deterministic model that takes input images, a target environment map and directly outputs the relighted 3D Gaussians.  We conducted both qualitative and quantitative comparisons with such a deterministic design (see Figure 5 and Table 4). While the deterministic model produces reasonable results, it consistently fails to generate sharp specular highlights(Figure-5), no matter how extensively it is trained. This limitation arises because deterministic models tend to over-smooth outputs when faced with ambiguities—such as estimating object roughness or specular highlights from sparse views—leading to suboptimal results.
>
> Additionally, this diffusion design has a few new interesting designs and capabilities.
>
> The concept of **“Virtual denoising views”**.Our appearance diffusion transformer uses a novel concept of "virtual denoising views." It iteratively denoise a set of virtual viewpoints under target lighting, and generates the relighted 3D Gaussians through this iterative process.  This set of “Virtual denoising views” is a new concept in our paper, and it can be arbitrary and different from input views. It can even change through the process of denoising. We analyzed the impact of varying the number of denoising views in Table 6(a).
>
> Interestingly, our model can be directly used for **other potential applications**: de-light, roughness editing.
> To support classifier-free guidance[1], we randomly remove the target environment maps by 10%, like token drop-out during training. And in inference, we can apply classifier-free guidance (cfg) with different weights. And a higher cfg weight mimics the editing effect of making objects more specular, and a small cfg weight mimics the editing effect of making objects more diffuse. Interestingly, setting the CFG weight to 0 (fully dropping the target environment map) achieves effects similarly to de-lighting, even though the model was not explicitly trained for de-lighting (see Figure 6).
>
> Finally, our method addresses a more challenging task than GS-LRM by enabling relightable reconstruction, whereas GS-LRM focuses solely on reconstruction. Our model learns to render self-shadows and specular highlights, which is non-trivial to learn without any shading priors. Moreover, we trained our model from scratch using a combination of diffusion loss and novel view rendering loss under target lighting conditions.
>
> [1] Ho, J., & Salimans, T. (2022). Classifier-free diffusion guidance. arXiv preprint arXiv:2207.12598.

---

> ### Author Response · Authors · 2024-11-20
> **First rebuttal For Reviewer Sueo. Part-3. Industrial applications**
>
> **Reviewer Sueo**:
> *“The method still takes 2 to 3 seconds to render. In contrast, the geometry and textures obtained from other methods can be rendered using many real-time rendering techniques. Moreover, in current industrial applications, it is challenging to abandon triangle meshes and textures. Therefore, this method cannot be considered a viable solution for 3D digital assets. However, if this approach could be extended to scene relighting, its applicability could be significantly broadened.”*
>
> We appreciate the reviewer's perspective on the potential industrial applications of our method. We would like to clarify the following points:
>
> Our model requires 2–3 seconds to produce relighted 3D Gaussians. However, once generated, these 3D Gaussians can be rendered efficiently from any viewpoint. For instance, the original 3D Gaussians paper [1] demonstrated rendering speeds exceeding 100 fps at 1080p resolution. Thus, for scenarios with relatively static lighting, the relighted 3D Gaussians do not need to be regenerated, and rendering efficiency is not a concern. We acknowledge that adopting 3D Gaussians directly for industry is not trivial.  And in the dynamic lighting case, we also agree with the Reviewer that our model cannot be directly applied due to efficiency issues. Potential solutions include distilling the output into explicit representations, such as triangle meshes and BRDFs, to enhance compatibility with existing workflows.
>
> [1] Kerbl, B., Kopanas, G., Leimkühler, T., & Drettakis, G. (2023). 3D Gaussian Splatting for Real-Time Radiance Field Rendering. ACM Trans. Graph., 42(4), 139-1.

---

> ### Author Response · Authors · 2024-11-22
> **First rebuttal For Reviewer Sueo. Part-4. Geometry evaluation. Appendix updated**
>
> *"Please provide quantitative evaluations of the extracted geometries."*
>
>
> We have updated the geometry evaluation results, which can be found in Appendix A.7 and Table 7.
>
> Quick summary:
>
> Our method achieves a Chamfer distance of 0.39 on the Stanford-ORB benchmark, outperforming all relighting baselines.
>
> When compared to pure 3D reconstruction methods, our approach ranks second according to Table 2 of the Stanford-ORB paper.
>
> It is worth noting that while our model is trained from scratch using a relighting loss and does not rely on any pretrained reconstructor, our geometry quality benefits from advancements in data-driven feedforward 3D reconstruction models, such as GS-LRM and LGM.
>
>
> For completeness, here is the content from Appendix A.7:
>
> We evaluate the geometry quality of our method on the Stanford-ORB dataset using our Res-512 model with six input views.  For evaluation, we filter out Gaussian points with opacity below $0.02$ and randomly sample 30,000 points. The Chamfer distance is computed against the scanned ground-truth mesh using the official Stanford-ORB script, with results scaled by $2 \times 10^3$, as per the benchmark protocol. Table below (Table-7 in the Appendix) summarizes the geometry and relighting results. Our method achieves the best geometry score while requiring only six input images and seconds of processing time, in contrast to other methods that rely on dense view inputs and require hours for final relighting.
>
> | Method              | Geometry (Chamfer Distance ↓) | Novel View Relighting (PSNR-H ↑) | Novel View Relighting (PSNR-L ↑) | # Input Views | Runtime     |
> |---------------------|--------------------------------|----------------------------------|----------------------------------|---------------|-------------|
> | **Ours**           | **0.39**                      | 24.67                           | 31.52                           | **6**         | **~2 seconds** |
> | Neural-PBIR         | 0.43                          | **26.01**                       | **33.26**                       | ~60           | ~60 hours   |
> | IllumiNeRF          | N/A                           | 25.56                           | 32.74                           | ~60           | hours   |
> | NVDIFFREC-MC        | 0.51                          | 24.43                           | 31.60                           | ~60           | hours   |
> | InvRender           | 0.44                          | 23.76                           | 30.83                           | ~60           | hours   |
> | NeRFactor           | 9.53                          | 23.54                           | 30.38                           | ~60           | hours   |
> | NVDIFFREC           | 0.62                          | 22.91                           | 29.72                           | ~60           | hours   |
> | PhySG               | 9.28                          | 21.81                           | 28.11                           | ~60           |  hours   |

---

> ### Author Response · Authors · 2024-11-25
> **Updated qualitative results on predicted Geometry**
>
> Hi, Reviewer Sueo,
>
> Thank you for your suggestion to evaluate the predicted geometry. I previously shared quantitative results on the Stanford-ORB benchmark, and now I would like to update you with qualitative results.
>
> For Stanford-ORB benchmark, we have uploaded all predicted geometries of our method in the supplementary materials. We use our Res-512 model with six input views to generate the predicted Gaussians. Points with opacity below 0.05 are filtered out, and 30,000 points are randomly sampled for each object for visualization.
>
> Specifically, for the four objects in Stanford-ORB highlighted in Figure 3 of our paper, please look for folders with name: **pepsi**, **gnome**, **pitcher**, and **cactus**.
>
> We will also release this full set of predicted geomtries in the future.

---

> > ### Comment · Reviewer_Sueo · 2024-11-29
> >
> > Thank you for the detailed responses, which have addressed my concerns. I don't have further questions.

---

### Official Review · Reviewer_kqbi · 2024-11-01

**Soundness:** 2
**Presentation:** 3
**Contribution:** 2
**Rating:** 8
**Confidence:** 5

**Summary:**

This paper tackles the task of reconstructing relightable 3D objects from sparse images. For this, the authors extend the idea of GS-LRM to incorporate the generation of a relightable appearance. Specifically, they feed the geometry feature from GS-LRM through a diffusion process based on the transformer architecture. The transformer attends to the target illumination and outputs appearance tokens. The combination of the appearance tokens from the newly added transformer and the geometry tokens from the original GS-LRM transformer concludes the generation of Gaussian Splats. Experiments on the Stanford-ORB and Objects-with-Lighting real-world benchmark as well as the TensoIR synthetic benchmark demonstrate the effectiveness of the proposed approach.

**Strengths:**

- originality-wise: I appreciate the proposed end-to-end transformer-based architecture for relightable 3D object reconstruction.
- quality-wise: when taking into the efficiency of the proposed approach, I think quantitatively the performance is good.
- clarity-wise: the paper is well-written and easy to follow.
- significance-wise: the task of relightable 3D object reconstruction is vital for many downstream applications, e.g., AR/VR and robotics.

**Weaknesses:**

## Concerns about the architecture

I think the authors missed an important question to answer: why do we choose the current architecture? Essentially, the mechanism of discarding many tokens (L231) is wasting the model's capability.

The root question is why do we need to use self-attention instead of cross-attention? Why do the environment maps and the denoised images need to be treated the same as the appearance tokens, especially since only the appearance tokens will be used for the Gaussian Splats rendering?

## Not enough understanding of the current model

Even though with the current architecture, I do not think the authors provide enough analysis. Specifically, have the authors visualized the attention maps of the transformer? What does the transformer learn? Does it attend to some specific areas in the environment map that cause the specular effects in the rendering? How does the transformer attend to those denoised images?

## Concerns about the qualitative results

In Fig. 3.(a), the produced Pepsi can's color is quite different from the ground truth. A similar thing happens to the gnome's colors. Further, the characters on the Pepsi can are quite blurry compared to NVDiffRec-MC / ground-truth. Additionally, in Fig. 3.(c), the RelitLRM produces quite different shows from the ground truth. However, the shadows are correctly predicted by both InvRender and TensoIR.

Why is this the case? Have the authors carefully studied the causes? Whether increasing the number of views will help? Will 16 views mitigate these issues as the authors state that "performance saturates around 16 views" (L525)? If even 16 views cannot resolve the issue, what are the intrinsic shortcomings of the proposed model?

I hope the authors can provide a systematic analysis for a good understanding of the model.

## Missed important baselines

I think the authors missed several quite related as well as important baselines, e.g., [a, b]. They both use the idea of diffusion-based relighting model to tackle the task of relightable reconstruction.

Especially IllumiNeRF [b], which directly tackles the task of relightable object reconstruction and competes on the benchmark of Stanford-ORB and TensoIR. Frankly speaking, [b] outperforms the proposed approach on both benchmarks quantitatively:
- PSNR-H / PSNR-L / SSIM / LPIPS: 24.67 / 31.52 / 0.969 / 0.032 (RelitLRM) vs 25.42 / 32.62 / 0.976 / 0.027 ([b] on Stanford-ORB)
- PSNR / SSIM / LPIPS: 29.05 / 0.936 / 0.082 (RelitLRM)) vs 29.709 / 0.947 / 0.072 ([b] on TensoIR)

[a] A Diffusion Approach to Radiance Field Relighting using Multi-Illumination Synthesis. EGSR 2024.

[b] IllumiNeRF: 3D Relighting without Inverse Rendering. ArXiv 2024.

## Concerns about the novelty claim

On L088, the authors claim the first contribution as "a regression-based geometry reconstructor followed by a diffusion-based appearance synthesizer". Though I appreciate the end-to-end transformer architecture, I may not be convinced that the idea is entirely novel since the above-mentioned IllumiNeRF has already proposed an almost exact idea. I would recommend modifying it to correctly position the work.

## Missed training details

What is the hardware setup required to train the model? How long does it take to complete the training, hours or days?

## Incorrect description about the benchmark

In L356, the authors state that the Stanford-ORB benchmark has "60 training views and 10 test views per lighting setup". This is not true. Please correct it.

## Typos

L290: "is involves" -> "involves"

**Questions:**

See "weakness".

---

> ### Author Response · Authors · 2024-11-20
> **First rebuttal For Reviewer kqbi - Part-1: "Concern about the architecture"**
>
> Due to the 5000-character limit per reply, I have split the rebuttal into multiple parts. This is Part 1 of the first round of rebuttal with Reviewer kqbi.
>
> We thank Reviewer kqbi for your detailed comments and valuable feedback. Below, we address your concerns.
>
> ## "Concern about the architecture"
> **Reviewer kqbi:**
> *“Why do the authors choose the current architecture? Why use self-attention instead of cross-attention? Why treat environment maps and denoised images the same as appearance tokens, given that only appearance tokens are used for Gaussian Splats rendering?”*
>
> Our choices were guided by two principles: simplicity and leveraging the success of large-scale vision and language models. Below, we elaborate on our rationale and provide experimental evidence to address your concerns.
>
> Our model handles input from three modalities: posed input images, target environment maps, and noisy target relighting views. A conventional approach might involve designing three modality-specific encoders (with or without shared parameters) followed by a decoder to aggregate features from all modalities and relight.  However, this approach introduces significant inductive biases and design complexity, such as how much parameter should we allocate for each modality encoder and the decoder.
>
> To prioritize simplicity, we adopted a lightweight modality-specific MLP to project tokens, followed by a shared stack of transformer layers using self-attention. This transformer processes all tokens jointly, and it would learn to extract modality-specific features and perform information aggregation simultaneously. Crucially, the transformer will learn by itself how to allocate capacity across modalities and tasks without manual tuning.
>
> Our approach draws inspiration from successful vision-language models like LLava [1, 2] and PaliGemma[3], which use CLIP to extract image features then project image features via lightweight MLPs and treat image and text tokens equivalently using transformer layers with self-attention. Similarly, our design treats all modality tokens equivalently after a lightweight MLP projector. This avoids complex inductive biases while enabling flexibility and scalability.
>
> **Reviewer kqbi:** *“Why do we use self-attention instead of cross attention”*
>
> If we use cross-attention between the appearance tokens and environment map tokens, it would constrain the number of learnable parameters for environment map tokens, potentially limiting their feature extraction capabilities. To address your concerns further, we conducted experiments comparing our self-attention-based design with a cross-attention counterpart.
>
> For the cross-attention counterpart, we replaced all transformer layers in the relighting transformer with cross-attention-based layers. Both models have the same amount of trainable parameters. We train it with 4 input views at Res-256 for 80k iterations, using the same setup, same data as our Res-256 model. We evaluate it on Stanford-ORB, Objects-with-Lighting and Held-out evaluation set, all with four input views at resolution of 256x256. Below are the results.
>
>
> **Stanford-ORB**
>
> | Method             | PSNR-H | PSNR-L | SSIM   | LPIPS  |
> |--------------------|--------|--------|--------|--------|
> | Cross-Attention    | 21.06  | 26.70  | 0.943  | 0.060  |
> | Self-Attention     | 22.97  | 29.42  | 0.967  | 0.0491 |
>
> **Objects-with-Lighting**
>
> | Method             | PSNR    | SSIM    | LPIPS  |
> |--------------------|---------|---------|--------|
> | Cross-Attention    | 13.464  | 0.491   | 0.564  |
> | Self-Attention     | 20.624  | 0.756   | 0.454  |
>
> **Held-Out Evaluation Set**
>
> | Method             | PSNR    | SSIM    | LPIPS  |
> |--------------------|---------|---------|--------|
> | Cross-Attention    | 26.34   | 0.906   | 0.078  |
> | Self-Attention     | 27.63   | 0.922   | 0.064  |
>
> We see the self-attention design outperforms the cross-attention design.We will include this comparison in the final version of our paper.
>
>
>
> [1]: Liu, H., Li, C., Wu, Q., & Lee, Y. J. (2024). Visual instruction tuning. Advances in neural information processing systems, 36.
>
> [2]: Liu, H., Li, C., Li, Y., & Lee, Y. J. (2024). Improved baselines with visual instruction tuning. In Proceedings of the IEEE/CVF Conference on Computer Vision and Pattern Recognition (pp. 26296-26306).
>
> [3]: Beyer, L., Steiner, A., Pinto, A. S., Kolesnikov, A., Wang, X., Salz, D., ... & Zhai, X. (2024). Paligemma: A versatile 3b vlm for transfer. arXiv preprint arXiv:2407.07726.

---

> ### Author Response · Authors · 2024-11-20
> **First rebuttal For Reviewer kqbi - Part-2: "Concern about the architecture" & "No enough understanding of the current model"**
>
> **Reviewer kqbi:**
> *“Essentially, the mechanism of discarding many tokens (L231) is wasting the model's capability.”*
>
> We want to add more details about the discarding operation:  After the final transformer block, we discard the tokens corresponding to the environment maps and noisy image patches. For a 4 input view, 4 noisy view setup with 256x256 resolutions. There are 4096 appearance tokens, 1024 noisy image tokens and 512 environment map tokens. We only kept the final 4096 appearance tokens to decode the color of each 3D Gaussian.
>
> Importantly, all transformer blocks in the diffusion model process all the tokens, and take part in aggregating features from environment map, noisy image, and appearance tokens throughout the network, contributing to the final Gaussian output. Thus, the model’s capacity is fully utilized during feature extraction and aggregation, ensuring no trainable parameters are wasted. Also as shown in the part-1 of the rebuttal, transformer with self-attention is an effective module for feature extraction and aggression.
>
> ## Not enough understanding of the current model
> **Reviewer kqbi:**
> “​​Specifically, have the authors visualized the attention maps of the transformer”, *What does the transformer learn? Does it attend to some specific areas in the environment map that cause the specular effects in the rendering? How does the transformer attend to those denoised images?*
>
> I want to first clarify that interpreting neural networks is extremely hard.
> The suggestion of visualizing attention maps does provide a viewpoint to peek into the mechanisms of the transformer, but still it cannot fully explain how the network learns to do relighting, moreover visualizing attention maps might not leads to correct interpretations, as pointed out by some literature [1, 2]
>
> We actually have visualized the attention map between appearance tokens and environment map tokens. Although we observed some patterns, they were not super consistent. Combined with the doubts surrounding the reliability of attention-based interpretations, we chose not to include these visualizations in the submission. We haven’t visualized it for noisy tokens.
>
> However, we are happy to provide visualization for both in the Appendix. I will update your with results in two days.
>
>
> [1] Serrano, S., & Smith, N. A. (2019, July). Is Attention Interpretable?. In Proceedings of the 57th Annual Meeting of the Association for Computational Linguistics (pp. 2931-2951).
>
> [2] Jain, S., & Wallace, B. C. (2019, June). Attention is not Explanation. In Proceedings of the 2019 Conference of the North American Chapter of the Association for Computational Linguistics: Human Language Technologies, Volume 1 (Long and Short Papers) (pp. 3543-3556).

---

> ### Author Response · Authors · 2024-11-20
> **First rebuttal For Reviewer kqbi - Part-3: "Concerns about the qualitative results"**
>
> Thanks for pointing out these reasonable concerns on our qualitative results. Here we address these issues individually.
>
> **Reviewer kqbi:**
> *"In Fig. 3.(a), the produced Pepsi can's color is quite different from the ground truth"*
>
> We believe the main reason for this is the scale ambiguities between unknown material albedo and unknown surrounding environment maps. This is known as a common issue in the domain of inverse rendering. Per-channel scaling helps alleviate this problem a little bit, we provide the visualization of the Pepsi after per-channel scaling in the updated Appendix (Figure-7).
>
> **Reviewer kqbi:**
> *Blurred characters on the Pepsi can:*
>
> The blurriness isn't a limitation of our method. We believe it might come from two potential problems: out-of-distribution camera field of view and noise in camera pose:
>
> 1. *Out-of-distribution camera intrinsics*. Our model was trained primarily on data rendered with a camera field of view (FOV) of 50 degrees.  However, Stanford-ORB dataset is captured with a camera field of view of  around 20.3 degrees. Which is significantly out of distribution for our model(especially the input channels for the Plucke-rays). In contrast, datasets like TensoIR-Synthetic and Objects-with-Lighting (after center cropping) have FOVs of 39.5 and 40.9 degrees, respectively, which are closer to our training distribution.
> 2. *Noise in camera pose*. Real-world captures have errors in camera pose, and this is particularly challenging when the input view is sparse. Dense views methods can mitigate the effects of such inaccuracies by averaging them out, but sparse views amplify these issues. Furthermore, our model was trained on synthetic data, which does not contain camera pose noise. We acknowledge that developing training methods robust to camera pose noise and distortion is an important avenue for future research.
>
> Nonetheless, our model demonstrates overall superior performance, particularly in rendering more accurate specular patterns compared to the baselines illustrated in Figure 3.
>
> **Reviewer kqbi:**
> *"Additionally, in Fig. 3.(c), the RelitLRM produces quite different shows from the ground truth. However, the shadows are correctly predicted by both InvRender and TensoIR."*
>
> For the Lego, we argue that our model produces a shadow that is closer to the ground truth than the baselines. The shadow in the ground-truth is very soft, but both InvRender and TensorIR cast very hard shadows, while our model does have soft shadows.
>
> For the hotdog, We acknowledge that the shadow cast by the hotdog onto the plate in our results is not as sharp or accurate as the TensoIR baseline. It is worth noting, TensoIR achieves this with ray marching to compute visibility mask and shadows, a task-specific inductive bias explicitly designed for shadow casting. In contrast, our method does not use any such inductive bias or designs, the model just learns by itself through stacks of transformer layers. Such a simple design can already produce quite reasonable shadows and highlights, and it’s already non-trivial.
>
> Additionally, when examining the bottom parts of the hotdogs in Fig. 3.(c), both InvRender and TensoIR struggle with shadow removal, producing overly dark results caused by shadows in the input images. While our methods perform significantly better in removing these input shadows. (I added the input images of our method for this “hotdog” in Figure-8 of the revised appendix, where you can see shadows in the bottom of the hotdog.)
>
> **Reviewer kqbi:**
> *“Whether increasing the number of views will help? Will 16 views mitigate these issues as the authors state that "performance saturates around 16 views" (L525)?"*
>
> While increasing the number of input views helps reduce ambiguities, certain material-illumination ambiguities are fundamentally irresolvable. Consider the classic 'white furnace test': for a Lambertian object, doubling the reflectance while halving the lighting intensity produces identical radiance. And images of the objects under multiple lightings can address such ambiguities, which is different from the problem setup we are addressing.
>
> We show the visual results of using 4, 8, 12 input views for the lego in the Figure-9 of the updated Appendix. Most of the improvements come from more accurate textures in regions covered insufficiently with fewer inputs. And I think more such visualization definitely helps reader in understanding our model, and we will provide more.

---

> ### Author Response · Authors · 2024-11-20
> **First rebuttal For Reviewer kqbi - Part-4: "Concerns about the qualitative results" & “Missed important baselines”.**
>
> **Reviewer kqbi:**
> *“what are the intrinsic shortcomings of the proposed model? I hope the authors can provide a systematic analysis for a good understanding of the model.”*
>
> Thank you for this insightful question. We agree that discussing the fundamental limitations of our method can provide a clearer understanding of its strengths and shortcomings.
>
> For traditional optimization based inverse rendering methods, most part of the framework is quite interpretable, allowing people to anticipate when it would work and when the method would fail or produce severe artifacts. For example,
> * Previous methods like TensoIR can produce sharp cast-shadows, but shadow removal and global illumination is hard for them.
> * Previous methods relying on mesh-based representations suffer from topology problems.
> * Very challenging to deal with highly specular objects.
> For our feedforward methods, I can identify a few fundamental limitations:
> * Since we use 3D Gaussians as the output representations, which inherently limits its ability to handle transparent objects.
> * We use spherical-harmonics up to order of 4 to represent view-dependent colors, this design struggles with highly specular objects and cannot represent non-symmetric view-dependent colors effectively.
> * Since our model takes a sparse view as input, it’s very challenging to deal with objects with high self-occlusions, because sparse view won’t have sufficient coverage for occluded regions.
> * Out of distribution data. Like all data-driven methods, our model’s performance degrades on out-of-distribution data. Addressing this would require better data curation and augmentation strategies to improve robustness.
>
> While our feedforward approach presents some fundamental limitations, it’s by design simple and flexible, and it already produces reasonable results. With continued scaling and refinement, we expect further improvements, especially in handling challenging scenarios such as sharp shadows and specular highlights.
>
> Meanwhile, we will include more visual results in the appendix in two days. We also plan to release all the results on three benchmarks: Stanford-ORB, Objects-with-Lighting and TensoIR-Synthetic, in a big zip file.
>
>
> ## “Missed important baselines”.
> **Reviewer kqbi:**
> *Missed baselines of [a], [b]. Comparison with IllumiNeRF[b] on Stanford-ORB and TensoIR-Synthetic*
>
> Thank you for pointing out these related works. We acknowledge the relevance of both [a] and [b], which utilize learned diffusion priors for 3D relighting. However there are some fundamental differences we want to clarify. **Both these two methods cannot handle sparse-view input settings**, as they need a dense capture input(or NeRF as input). Moreover,  **both of them employ an optimization-based approach for 3D relighting which takes hours.**
>
> In contrast, our method addresses **feedforward sparse-view 3D relighting**, which produces 3D relighted results end-to-end in seconds, without using any optimization based approach.
>
> We appreciate the suggestion to include IllumiNeRF [b] in our main result table for completeness(Added in Table-1, and Table-2). While [b] achieves competitive results, it actually didn't outperform other optimization-based baselines already included in our table. (In Stanford-ORB, it underperforms Neural-PBIR already in our table. In TensoIR-Synthetic, it matches the results of TensoIR). We already added them in Table-1,2 of our updated version.
>
> We want to clarify that we compare with dense-view optimization based approaches not because it’s fair to compare them. We compare with them for two reasons:
> 1. Lack of good baselines in feed-forward sparse-view 3D relighting methods.
> 2. Context for performance. These comparisons provide insight into how close our sparse-view method is to state-of-the-art dense-view approaches.
>
> One interesting detail to note is that IllumiNeRF's evaluation on TensoIR-Synthetic applies pixel-space scaling instead of in albedo space. When both applied rescaling in pixel-space,the Single-GPU results of IllumiNerf actually are worse than the original TensoIR which does not use such image-based diffusion relighting priors! For the 16-GPU setup (2.5 hour runtime with 16-A100 GPUS), IllumiNeRF’s results (PSNR of 29.709, LPIPS of 0.072)  matches the results of TensoIR (PSNR of 29.64, LPIPS of 0.068.).
>
> [a] A Diffusion Approach to Radiance Field Relighting using Multi-Illumination Synthesis. EGSR 2024.
>
> [b] IllumiNeRF: 3D Relighting without Inverse Rendering. NeurIPS 2024.

---

> ### Author Response · Authors · 2024-11-20
> **First rebuttal For Reviewer kqbi - Part-5: "Concerns about novelty " & "Training details" & "Description of the benchmark" & Typos**
>
> ## Concerns about the novelty claim
> **Reviewer kqbi:**
> *“I may not be convinced that the idea is entirely novel since the above-mentioned IllumiNeRF[b] has already proposed an almost exact idea”.*
>
> We respectfully disagree with Reviewer kqbi’s assessment. While our work and IllumiNeRF share some conceptual similarities: using diffusion prior for relighting, there are fundamental differences in problem settings, methodology, and capabilities that distinguish our approach.
>
> 1. Problem settings:
> IllumiNeRF addresses dense-view 3D relighting, which assumes the availability of dense input views and is not designed to handle sparse-view scenarios.
>
> Our work, on the other hand, tackles the more challenging task of sparse-view relightable 3D reconstruction, enabling effective reconstruction and relighting with significantly fewer input views.
>
> 2. Method difference.
>
> At high level, 	IllumiNeRF relies on an optimization-based framework, requiring hours to relight the object under the target environment map. In contrast, our method employs a feedforward architecture that bypasses the need for time-consuming optimization, producing relighted 3D Gaussians in seconds during inference.
>
> In detail, IllumiNeRF uses an image-based relighting diffusion model to relight each image independently, and use optimization-based approach to distill relighted dense view images to a latent NeRF representation.
> Our method is a feedforward model trained from scratch. During inference it avoids cumbersome intermediate steps and enables fast and efficient relighting.
>
> [b] IllumiNeRF: 3D Relighting without Inverse Rendering. NeurIPS 2024.
>
>
> ## Missed training details
> **Reviewer kqbi:**
> *“What is the hardware setup required to train the model? How long does it take to complete the training, hours or days?”.  *
>
> Thanks for pointing this out. This question is raised by all the reviewers, we already add them in the updated Appendix (A.4).
>
> We train our model with 32 A100GPUs(40G VRAM). For the Res-256 Model, we train for 4 days, and for the Res512 model, we finetune for another 2 days.
>
> ## Incorrect description about the benchmark
> **Reviewer kqbi:**
> *“In L356, the authors state that the Stanford-ORB benchmark has "60 training views and 10 test views per lighting setup". This is not true. Please correct it”.  *
>
> “60 training views and 10 test views per lighting setup” in our paper comes from the second paragraph of section 3.4.3 of Stanford-ORB paper: “We take images from approximately 70 viewpoints roughly uniformly covering 360° views of the objects, including 10 test views and 60 training views”.
> We will change our sentences slightly to:  “approximately 60 training views and 10 test views per lighting setup per object”.  If this is not accurate enough, can you also elaborate more on this?
>
> ## Typos
> Thanks for the detailed reading, fixed it.

---

> ### Comment · Reviewer_kqbi · 2024-11-22
>
> **I apologize for the messed up replies as I think that replying to individual blocks will create a separate thread. I now aggregated all my replies here and deleted others.**
>
> Thank you for adding more clarifications and running new experiments. I appreciate them a lot.
>
> **1. Regarding self-attention vs cross-attention**
>
> For the self-attention vs cross-attention comparison, the results on Objects-with-Lighting are quite different from those on Stanford-ORB and Held-Out Evaluation Set. Specifically, the performances of two different attention mechanisms on Stanford-ORB and the Held-Out Evaluation Set are close while those on the Objects-with-Lighting are dramatically different. Why is this the case? Can authors provide some insights?
>
> **2. Regarding the rendering rescaling**
>
> Can the authors clarify whether the quantitative results reported in both Tab. 1 and Tab. 2 are after the rescaling or before the rescaling? I am confused now.
>
> **3. About qualitative results of using more views in Fig. 9**
>
> Can authors add all 12 source views?
>
> I also notice that with more views added, the renderings gradually become brighter and brighter, similar to the effect of over-exposure. Can authors provide some insights on why with a **fixed** environment map and various number of source views, the final relighting appearance changes a lot? I think this is a different issue with missing details with less views.
>
> **4. Regarding the novelty**
>
> I appreciate the authors' clarifications.
>
> I never underestimate the contributions from 1) the sparse view; and 2) the feedforward manner of the proposed RelitLRM.
>
> However, the first contribution in the paper (L088 - 092) is the following:
>
> > Novel transformer-based generative relighting architecture. We propose to use a regression-based geometry reconstructor followed by a diffusion-based appearance synthesizer (both modeled by transformers and trained end-to-end), to disentangle geometry from appearance, allowing better modeling of the uncertainty in relighting.
>
> My statement is that the idea of `a regression-based geometry reconstructor followed by a diffusion-based appearance synthesizer ... allowing better modeling of the uncertainty in relighting` is not completely new as it has been proposed in other works.
>
> **I only suggest correctly positioning the work.**
>
> **5. Regarding the dataset description**
>
> Adding the word "approximation" sounds good to me. I downloaded the Stanford-ORB data and even after briefly reviewing it up to the 2nd scene, i.e., baking_scene002, I found it does not have exactly 60 training views.
>
> I want the description to be precise. As you said, you wrote the description based on the original Stanford-ORB paper. What if other readers write their own papers by referring to the incorrect description?

---

> > ### Author Response · Authors · 2024-11-27
> > **Second round of Rebuttal for Reviewer kqbi. Part-2**
> >
> > **3. About qualitative results of using more views in Fig. 9**
> >
> > Thank you for highlighting this interesting phenomenon. After analyzing more samples, we observed that adding more views often—but not always—results in increased output brightness. To investigate, we conducted an very interesting experiment, which we believe reveals the primary cause.
> >
> > To explain the experiment, I want to first recap some of the details of our model:
> > For each input images, we predict pixel-aligned 3D Gaussians for each pixel, and we directly concate predicted 3D Gaussians from all input views together as the final predictions!  So, when more input views are provided, there would be more and more predicted Gaussians in overlapping regions between viewpoints.
> >
> > Experiment:
> > We feed the lego scene again to the model under the same 8-view input setting but modified the final predictions to include Gaussians from only the first four input views. Note that we didn't change the forward pass of the transformers. The rendering results are shown in Figure-12 (second-last column).
> >
> > Key observations:
> > In overlapping regions between views, some Gaussians appear masked out. However, remember that we predict a Gaussian at each pixel, and these white regions are indeed visible to the first four viewpoints,  these Gaussians in the white region is not explicitly masked out, but learned a low opacity scores. We also visualize the predicted Gaussians from the first 1, 2, 3 input views to better understand this effect(still the 8 input view setting.) in Figure-12.
> >
> > Our conjecture:
> > We conjecture that with more and more input views, there will be more and more floaters with low opacity around the objects, especially in overlapping regions between input views. And these floaters makes the object appear brighter and brighter. We also want to clarify that these floaters might not be meaningless, they should have learned to also add some finner details to the final results.
> >
> > To validate this hypothesis, we visualized the results after pruning Gaussians with opacity scores below 0.25. For 8-view and 12-view inputs, this reduced the object’s brightness while maintaining visually high-quality renderings, as in the updated version of the Figure-9 (last two columns of the bottom half of the figure). However, pruning also removed some high-frequency details, suggesting that these low-opacity floaters are a mechanism learned by the model to enhance detail with more input views, with a side-effect of changing object's brightness.
> >
> > We hope this explanation clarifies the observed phenomenon. This observation also highlights a key challenge for scaling feed-forward 3D models like GS-LRM and RelitLRM to handle a large number of input views (e.g., over 100). Improved methods for aggregating predictions across viewpoints, or entirely new paradigms, may be required to address this issue effectively.
> >
> > We have updated the figure-9 to include the full set of 12 input images.
> >
> > **4. Regarding the novelty.**
> >
> > I appreciate reviewer's comments and clarification.
> >
> > I change the L088-092 to better position our work:
> >
> > **Novel feed-forward generative 3D relighting architecture.** We end-to-end learn a generative 3D relightable reconstruction model with deterministic geometry reconstruction and probabilistic radiance generation. Our approach bypasses the explicit appearance decomposition and shading, and directly generates relighted radiance, producing realistic view-dependents appearances and self-shadows.
> >
> > With a focus on end-to-end feed-forward 3D relightable reconstruction.
> >
> > **5. Regarding the dataset description**
> >
> > I agree with Reviewer kqbi and appreciate your help on improving our submission. Precise and Correct description is important.

---

> ### Author Response · Authors · 2024-11-22
> **First rebuttal For Reviewer kqbi - Part-6: "Attention Map Visualization". Appendix and Supplementary files updated**
>
> **Reviewer kqbi:**
> *"Specifically, have the authors visualized the attention maps of the transformer? What does the transformer learn? Does it attend to some specific areas in the environment map that cause the specular effects in the rendering? "*
>
> I appreciate your patience, and help on making this submission better.
>
> I want to update you about the visualization of attention maps between appearance tokens and environment maps. I put details of them in the Appendix A.8 and Figure 10, 11 for visualizations.  Please refer it for details. Also,  we put more video results in the Supplementary. Please see attention_map_visualization folder.  In the supplementary files, we visualized the attention map at the first layer and the last layer, for four objects. We showed the results in video format: where first-row shows two attention maps, corresponding to the center patch of the first and second input image.
> The second row shows input image.
> The last rows, shows the environment map (which is rotating horizontally), we showed tow tone-mapping of the environment map for visualization.
>
>
> Below, I copy text in the A.8 in Appendix here
>
> We would like to provide updates regarding the visualization of attention maps between appearance tokens and environment map tokens. Detailed explanations are provided in Appendix A.8, with visualizations in Figures 10 and 11. Additionally, more video results can be found in the Supplementary Material under the attention_map_visualization folder.
>
> In the supplementary files, we visualize the attention maps from both the first and last transformer layers for four objects. The results are presented as videos:
> * First row: Attention maps corresponding to the center patch of the first and second input images.
> * Second row: Input images.
> * Last rows: Environment maps two different tone-mapping styles.
>
>
> For completeness, the text from Appendix A.8 is copied below:
>
> Each transformer layer employs multi-head self-attention with 16 heads, each of dimension 64, resulting in a total hidden dimension of 1024. For visualization, we concatenate the key and value vectors from all heads and compute attention using the aggregated keys and values.
>
> We use our Res-256 model at four-view input setup. for visualization. We show visualized results for two objects, each with two input lighting setups and two target lightings. For each input setup, we visualize the attention map for the image patch at the center of the first and second input image. For each target lighting, we horizontally shift it by $1/4, 1/2, 3/4$ to create four target environment map.   For each target lighting, we horizontally shift the environment map by $1/4$, $1/2$, and $3/4$, creating four variations per lighting. In total, we visualize $2 \times 2 \times 2 \times 4 = 32$ attention maps per object. The results are shown in Figures, 10, 11.
>
>
>
> For the visualized attention, we summarize some empirical patterns, though not super consistent.
>   1. **Lighting rotation consistency**. The attention map between appearance token and environment map tokens is quite consistent to rotations of the environment map. Note that environment map is tokenized by concating the color and ray direction for each pixel. This consistency implies a strong dependence on directional cues in environment map.
>   2. **Input lighting stability**. While less consistent than rotation, attention maps are relatively stable across changes in input lighting. This might suggest, that the attention map learned something about the appearance models of the objects.
>   3. Contrary to intuition, attention maps for specular objects (Figure 10}) are not noticeably sharper or sparser compared to those for diffuse objects, as seen in Figure~11.
>   4.  Highlights in attention maps do not consistently align with directions where target lighting most affects object appearance. For instance, in Figure-10 (first row), appearance tokens corresponding to the object’s bottom aggregate lighting from the top of the environment map, contrary to expectation.

---

> ### Author Response · Authors · 2024-11-27
> **Second round of Rebuttal for Reviewer kqbi. Part-1**
>
> We appreciate Reviewer kqbi's new comments a lot, here we address your questions.  Due to word limit, we split the response to two part.
>
>
> **1. Regarding self-attention vs cross-attention**
>
> Thank you for highlighting this question regarding the performance differences on the Objects-with-Lighting benchmark.
>
> I looked at the results of Cross-Attention model on Objects-with-Lighting.  I observed that it often produces overly dark predictions. This suggests that the Cross-Attention model is less robust, possibly because the cross-attention mechanism lacks sufficient learnable parameters and compute to extract robust features from the environment maps. Specifically, it uses only a small MLP to extract per-patch features before performing cross-attention with appearance tokens.
>
> Additionally, we think there are several potential reasons related to the benchmark itself as well:
>
> 1. The Objects-with-Lighting benchmark contains only 42 test images (7 objects × 3 test views × 2 lightings). In contrast, Stanford-ORB includes ~400 test images (14 objects × ~10 views × 3 lightings), TensoIR-Synthetic has 4,000 test images (4 objects × 5 lightings × 200 views), and our Held-Out Evaluation Set contains 390 test images (13 objects × 5 lightings × 6 views). The significantly smaller size of the Objects-with-Lighting dataset introduces higher variance in evaluation results.
> 2. Systematic Differences in Benchmark. Some methods exhibit significant performance drops on the Objects-with-Lighting benchmark. For example, NVDIFFREC-MC ranks 3rd on Stanford-ORB but falls to last place (8th) on Objects-with-Lighting in Table 2. Additionally, SSIM scores across all methods on Objects-with-Lighting (mostly around 0.75, below 0.85) are much lower compared to Stanford-ORB (>0.95) and TensoIR-Synthetic (>0.9), indicating some systematic differences in the Objects-with-Lighting benchmark.
> 3. Performance gaps and rankings between methods are not always consistent across benchmarks. For example, NVDIFFREC-MC outperforms InvRender by 0.7 PSNR-L on Stanford-ORB but underperforms by 3.2 PSNR-L on Objects-with-Lighting; InvRender underperforms TensoIR by 0.7 PSNR-L on Objects-with-Lighting but by over 4.6 PSNR-L on TensoIR-Synthetic. This inconsistency seems to be a general problems for all these relighting benchmarks, and migh deserve some discussions. One potential reason for this might be that all these benchmarks contain relatively few objects (around 10).  Another potential reason is that Relighting is inherently ambiguous—multiple relighting results can be considered correct for a single scene. However, deterministic metrics like PSNR may not accurately capture this ambiguity, and leads to skewed evaluations.
>
> To provide a more complete comparison between the Self-Attention and Cross-Attention models, we have added evaluations on the TensoIR-Synthetic benchmark and included results for six-view input versions on both Objects-with-Lighting and Stanford-ORB. Below are results:
>
>
> ### Stanford-ORB. 6-View Input
>
> | Method             | PSNR-H | PSNR-L | SSIM   | LPIPS  |
> |--------------------|--------|--------|--------|--------|
> | Cross-Attention    | 21.28  |  27.00 |  0.947 | 0.059  |
> | Self-Attention     | 23.35 | 29.87   | 0.968  | 0.051  |
>
> ### Objects-with-Lighting. 6-View input
>
> | Method             | PSNR    | SSIM    | LPIPS  |
> |--------------------|---------|---------|--------|
> | Cross-Attention    | 14.09   |  0.511  |  0.543 |
> | Self-Attention     | 21.13   |  0.761  |  0.450 |
>
> ### TensoIR-Synthetic. 8-View input
>
> | Method             | PSNR    | SSIM    | LPIPS  |
> |--------------------|---------|---------|--------|
> | Cross-Attention    | 21.78   | 0.903   | 0.117  |
> | Self-Attention     | 22.91   | 0.908   | 0.115  |
>
> **2. Regarding the rendering rescaling**
>
> Sorry for the confusion.
>
> All quantitative evaluations involve rescaling, but no visualizations in the main paper are rescaled.
>
> For evaluations on the Stanford-ORB, Objects-with-Lighting, and TensoIR-Synthetic benchmarks, we generate all relit images without rescaling and use the official evaluation scripts provided by these benchmarks. Rescaling is performed within each benchmark’s evaluation code. More specifically, there are two different types of rescaling:
> *per-image rescaling*, applied in  Stanford-ORB and Objects-with-lighting.
> *per-scene rescaling*, applied by TensoIR-Synthetic, where all results in each scene under target lighting shares the same rescaling factor.
>
> We updated the experiment section in the paper to clarify this process. ( see the last sentence of figure-3's caption and line-372.)

---

> ### Comment · Reviewer_kqbi · 2024-11-27
>
> I thank the authors' efforts and new experiments. They are interesting and important. Please incorporate the discussions here in the final paper as they will make it more solid and insightful.
>
> Most of my concerns have been addressed, so I raised my score and leaned toward acceptance.
>
> Some minor things: I noticed that the newly added IllumiNeRF results were not referred to in the reference, please add them. Further, I would suggest adding some discussions about [a] and [b] in the related work to provide more context.
>
> [a] A Diffusion Approach to Radiance Field Relighting using Multi-Illumination Synthesis. EGSR 2024.
>
> [b] IllumiNeRF: 3D Relighting without Inverse Rendering. ArXiv 2024.

---

> > ### Author Response · Authors · 2024-11-28
> > **Reply to Reviewer kqbi**
> >
> > We thank reviewer kqbi for the engagement and insightful questions, which have helped improve our paper.
> >
> > We will incorporate the interesting discussions from the rebuttal into future revisions of the paper. Additionally, we will reference both suggested works and include a description of their relevance in the related work section.

---

### Official Review · Reviewer_GDsv · 2024-11-02

**Soundness:** 3
**Presentation:** 3
**Contribution:** 3
**Rating:** 8
**Confidence:** 3

**Summary:**

The paper presents a method to generate a relighable 3D representation using 3DGS for the geometry and a diffusion based model to get the illumination dependent appearance parameters for those Gaussians.
The geometry is predicted in form of per pixel Gaussians from the sparse input views (4 - 16) in a single Transformer forward step. The tokens of the geometric representation is concatenated with HDR features extracted from the illumination given as environment map and the noise target views. After denoising the tokens for everything except the input gaussians are discarded. The remaining tokens are decoded into the appearance (SH) of the Gaussians. The Gaussians are then rasterized into any novel view. The diffusion model is trained such that the lighting of the rendered Gaussians should match the lighting of the input environment map. During inference this environment map and the target view camera pose can be arbitrarily be chosen, thus a scene can be reconstructed from sparse views and then relit.

**Strengths:**

* Pseudocode in A.1clarifies all misunderstandings from the text descriptions. I can not stress enough how much I like this way of communicating the pipeline.
* The approach uses a diffusion process instead of a multi-view optimization, which is exceptionally fast in direct comparison.
* Trained on a rich dataset with a huge variety of lighting conditions.
* Light representation (input) is a simple environment map. Changing lighting is therefore very simple (no complicated neural light representation)

**Weaknesses:**

* Knowing only even a fraction of the research and work that has gone into disentangling the ambiguity between lighting and shading it rubs me the wrong way to read something that suggests it solved it without actually addressing the core problem. The method does not really decompose a scene into geometry, lighting and shading and is not usable if the use case would require extracting or editing reflectance properties. The way I see it this paper does relighting by decomposing a scene into geometry and appearance, however this is very different to what methods which explicitly extract reflectance and illumination do. The problem statement is profoundly different if you have to produce a explicit, reflectance representation which represents some underlying physical property of the surface, compared to just estimating the product of light and reflectance. I don't think much has to be changed to show respect for this difference: In 2.1 it should be mentioned that the method models the appearance under arbitrary illumination without an explicit reflectance representation. In the introduction the claim that the ambiguity between shading and lighting, is overcome should be phrased more carefully or be clarified. As far as I understand this paper estimates appearance as the integrated product between the unknown shading and a given lighting in form of a view dependent SH. This is really great work, but should not be confused with reflectance decomposition.

**Questions:**

* In the tables with the numbers for metrics please highlight the best numbers (bold)
* What hardware was used for training the model? Training time, memory requirements. Add to A.4
* In theory the method should work for objects that traditionally have challenging reflectance properties such as hair or fur. I am not sure if hair and fur were part of the training dataset, but it still might be interesting to see if it works.

---

> ### Author Response · Authors · 2024-11-20
> **First Rebuttal for Reviewer GDsv**
>
> Thank you very much for your thoughtful feedback.  We are greatly encouraged by your acknowledgement of our method’s efficiency, flexibility and our paper’s clarity.
>
> Regarding the concerns you raised about reflectance decomposition, we fully acknowledge the difference between our method and traditional approaches aimed at decomposing a scene into geometry, explicit reflectance (e.g., BRDF), and lighting components.  We aim to bypass this inverse process, bypass the explicit appearances and shading process, and directly generates the relighted appearances on the objects. This merged pipeline learns to handle the ambiguity in this problem by itself, through tremendous amounts of data.
>
> There are pros and cons. On the positive side, it allows us to handle uncertainty effectively, achieving robust performance with sparse-view inputs and significantly faster processing. However, as you rightly noted, our method does not offer an explicit appearance model, which may limit its utility for material editing and similar applications.
>
> To address these points more clearly in the paper, we have rephrased the last sentence of Section 2.1 to: “Our model bypasses explicit appearance decomposition and shading, directly generates relighted radiance, enabling high-quality relighting and rendering under unknown lighting conditions with sparse views, and offering advantages in scalability and practicality.”
> And we have added a sentence in the Introduction:“ Unlike traditional inverse rendering techniques that explicitly decompose appearance and shading, RelitLRM introduces an end-to-end relighting model directly controlled by environment maps.” in the second paragraph.
>
> Additionally, we can also update the limitations section to note that our approach does not support material editing due to the lack of an explicit appearance decomposition.
>
> For your other questions:
>
> **“In the tables with the numbers for metrics please highlight the best numbers (bold)”**
>
> Thanks for the suggestion, I bold the best metrics in all the tables. Additionally, for the dense optimization-based methods, we have ranked the results based on PSNR, listing them in descending order from top to bottom. For example, in Table 1, the optimization-based methods are ranked according to PSNR-H on the Stanford-ORB dataset.
>
>
> **“What hardware was used for training the model? Training time, memory requirements. Add to A.4”**
>
> Thanks for pointing out this question. I added. Here is the answer: We train our model with 32 A100GPUs(40G VRAM). For the Res-256 Model, we train for 4 days, and for the Res512 model, we finetune for another 2 days.
>
> **"In theory the method should work for objects that traditionally have challenging reflectance properties such as hair or fur. I am not sure if hair and fur were part of the training dataset, but it still might be interesting to see if it works."**
>
> I think this is a very interesting question. Due to time limit, I haven't finished experimenting with it yet, I will update you with this results in two days!

---

> ### Author Response · Authors · 2024-11-22
> **First Rebuttal for Reviewer GDsv - Part-2  Results for hair and fur.  Supplementary file updated**
>
> *"In theory the method should work for objects that traditionally have challenging reflectance properties such as hair or fur. I am not sure if hair and fur were part of the training dataset, but it still might be interesting to see if it works."*
>
>
> Thank you for your feedback and patience. I would like to update you with results on challenging objects like hair and fur.
>
>  I test three objects in this challenging category: hair, fur and cloth, for each of them I render with five environment maps, (same environment map as TensoIR-Synthetic benchmarks).
> I tested them using our Res512 model, and I show relighting on four target lighting for each of these objects.
>
> The results are provided as video visualizations in the updated supplementary files (not in the Appendix). For each object, we include two types of videos:
>
>  * light_rotating_videos:   First row of the video:  Relighted results across six viewpoints, with the first two matching the input viewpoints;  Second row: input images.;Last row: tTarget environment maps, visualized with two different tone-mapped formats.
> * camera_rotating_videos: Novel view renderings of the relit objects.. Bottom row shows input images.
>
> From the videos, it is evident that hair and fur remain extremely challenging, and I observe two major type of artifacts:
>  1. Sharp specular highlights in hair cannot be accurately reproduced
>  2. Subtle scattering effects are poorly captured.
>
>  I can think of two potential reasons for this:
>  1. Sparse View Coverage: Hair and fur involve intricate self-occlusions and fine details that are difficult to fully capture with sparse views.
>  2. Training Data Limitations: These high-quality objects likely represent a small fraction of the training data. They are significantly higher in quality than most assets in Objaverse, the primary source of our training data. (For a quick visualization of the Objaverse dataset, you can explore it here: https://objaverse.allenai.org/explore)

---

> > ### Comment · Reviewer_GDsv · 2024-11-29
> >
> > Thank you for the clarifications and additional experiments.
> > I unfortunately was not able to contribute more actively to the discussion for private reasons and I am sorry for that. You addressed my concerns and the questions of other reviewers really well. I already liked your approach before the rebuttal but with the added clarifications from the rebuttals I can embrace it wholeheartedly.
> > In my opinion this paper should clearly be accepted an I will update my rating accordingly.

---

### Meta-Review · Area_Chair_zUr4 · 2024-12-14

**Metareview:**

The paper introduces a method to generate 3D gaussian representations from sparse views under novel lighting conditions. The paper was well-received by all reviewers and converged to all-positive scores, recommending acceptance. The reviewers highlighted the strong qualitative results and the effectiveness of the feed-forward paradigm.
I agree with the reviewers and follow their suggestion.

**Additional Comments On Reviewer Discussion:**

Pre-rebuttal, concerns were raised regarding missing comparisons and unclear description of the architecture and novelty. Further questions were raised regarding robustness to different number of input views. The paper was heavily discussed between reviewers and authors and in the end, the authors convinced all reviewers by addressing their concerns.

---

### Decision · Program_Chairs · 2025-01-22

Accept (Spotlight)